

# Vertical distribution of aerosol optical properties in the Po Valley during the 2012 summer campaigns

Silvia Bucci[1,2], Paolo Cristofanelli[3], Stefano Decesari[3], Angela Marinoni[3], Silvia Sandrini[3], Johannes Größ[4], Alfred Wiedensohler[4], Chiara F. Di Marco[5], Eiko Nemitz[5], Francesco Cairo[1], Luca Di Liberto[1], and Federico Fierli[1]

[1]Institute for Atmospheric Sciences and Climate of the National Research Council , (ISAC-CNR), Rome, Italy
[2]Sc Dept. of Physics, Ferrara University, Ferrara, Italy
[3]Institute for Atmospheric Sciences and Climate of the National Research Council, (ISAC-CNR), Bologna, Italy
[4]Leibniz Institute for Tropospheric Research, Leipzig, Germany
[5]Natural Environment Research Council, Centre for Ecology & Hydrology, Penicuik, UK

*Correspondence to:* Federico Fierli (f.fierli@isac.cnr.it), Silvia Bucci (s.bucci@isac.cnr.it)

**Abstract.** Studying the vertical distribution of aerosol particles physical and chemical properties in the troposphere is essential to understand the relative importance of local emissions processes vs. long-range transport on column-integrated aerosol properties (e.g. the aerosol optical depth (AOD), affecting regional climate) as well as on the aerosol burden and its impacts on air quality at the ground. The main objective of this paper is to investigate the transport of desert dust in the middle troposphere

and its intrusion into the planetary boundary layer (PBL) over the Po Valley (Italy), a region considered one of the major European pollution hot-spots for the frequency of particulate matter (PM) limit values exceedances. Events of mineral aerosol uplift from local (soil) sources and phenomena of hygroscopic growth at the ground are also investigated, possibly affecting the PM concentration in the region as well. During the PEGASOS 2012 field campaign, an integrated observing-modeling system was set up based on near-surface measurements (particle concentration and chemistry), vertical profiling (backscatter

coefficient profiles from LiDAR and radiosoundings) and Lagrangian air masses transport simulations by FLEXPART model. Measurements were taken at the San Pietro Capofiume supersite (44°39' N, 11°37' 0 E, 11 m a.s.l), located in a rural area relatively close to some major urban and industrial emissive areas in the Po Valley. Mt. Cimone (44°12' N, 10°42' E, 2165 m a.s.l.) WMO/GAW station observations are also included in the study to characterize regional-scale variability. Results show that, in the Po Valley, aerosol is detected mainly below 2000 m a.s.l. with a prevalent occurrence of non-depolarizing particles

(> 50 % throughout the campaign) and a vertical distribution modulated by the PBL daily evolution. Two intense events of mineral dust transport from Northern Africa (19–21 June and 29 June to 2 July) are observed, with layers advected mainly above 2000 m height, but lately sinking and mixing in the PBL. As a consequence, a non-negligible occurrence of mineral dust is observed close to the ground (∼ 7 % of occurrence during a 1-month campaign). The observations unambiguously show Saharan dust layers intruding the Po Valley summertime mixing layer and directly affecting the aerosol concentrations near the

surface. Finally, LiDAR observations indicate also strong variability in aerosol on shorter timescales (hourly) highlighting: a) events of hygroscopic growth of anthropogenic aerosol, visible in shallow layers of low depolarization near the ground. Such events are identified during early morning hours at high relative humidity (RH) conditions (RH > 80 %). The process is ob-



served concurrently with high PM1 nitrate concentration (up to 15 µg cm$^{-3}$), hence mainly explicable by deliquescence of fine anthropogenic particles, and during mineral dust intrusion episodes, when water condensation on dust particles could instead represent the dominant contribution; b) frequent events (mean diurnal occurrence of ∼22% during the whole campaign) of rapid uplift of mineral depolarizing particles in afternoon-evening hours up to 2000 m a.s.l. height. The origin of such particles cannot be directly related to long-range transport events, being instead likely linked to processes of soil particles resuspension from agricultural lands.

## 1 Introduction

The Po river basin in Northern Italy is one of the most important emissive area in Europe, characterized by high concentration of both natural and anthropogenic aerosol and trace gases (Monks et al., 2009). The geographical location of this region, surrounded by two mountain ranges, promotes frequent occurrence of stagnant meteorological conditions (Rossa et al., 2012), with accumulation of local pollution (Hanke et al., 2003; Crosier et al., 2007; Saarikoski et al., 2012; Decesari et al., 2014) from industrial, urban and agricultural emissions, and complex processes of aerosol-chemicals transformation. Such combination leads to unusually high concentrations of atmospheric pollutants and particulate matter, with frequent and prolonged periods of intense pollution. The large amount of people living in the region (more than 20M potentially exposed to high pollution levels) accentuates the need of accurate studies on particulate variability over the Po Valley. Observations show that the spatial distribution and composition of aerosol over the area is often complex and variable in space and time, due to the interplay of sources and transport regimes (Lelieveld et al., 2002). The co-emission of particles and NOx from combustion emissions and the widespread sources of ammonia from agricultural activities (Clarisse et al., 2009) lead to the accumulation of primary carbonaceous particles and secondary inorganic aerosols (ammonium nitrate) in the lower layers of the atmosphere. Crosier et al. (2007), during a campaign in summer 2004, observed that, under easterly flow, ammonium sulphate and organics dominated the sub-micron aerosol fraction while, under westerly anticyclonic flow, the large NOx and ammonium emissions at the surface resulted in a large ammonium nitrate concentration in air masses recirculating over the Po Valley. During summer 2012, under similar meteorological features (anticyclonic conditions and PBL air recirculation), Sandrini et al. (2016) observed a significant enhancement of secondary organic and inorganic aerosol mass. They also pointed out differences in rural and urban area aerosol behaviour: rural area, being characterized during night by higher relative humidity and lower temperature compared to the urban one, showed higher fine nitrate nocturnal concentration and formation of ammonium nitrate in large accumulation mode aerosol (0.42-1.2 $\mu m$). This large amount of highly hydrophilic compounds significantly increase particles light scattering due to additional water uptake: ammonium nitrate-rich particles are very hydrophilic and, in conjunction with the high relative humidity conditions often encountered in the Po Valley floor, contribute to the build-up of hazes in the region, also in the summer season (Hodas et al., 2014). At the same time, the relative proximity to the Sahara desert, that represents the major mineral dust source of the planet (Prospero et al., 2002; Washington et al., 2003), makes this region subject to long range mineral dust transport, especially during the summer season (Kalivitis et al., 2007; Marinoni et al., 2008; Pederzoli et al., 2010; Carnevale et al., 2012). Saharan dust particles are uplifted to the mid-troposphere levels (up to 5 km) by the strong surface winds





and the large-scale convection that typically involves north Sahara during the summer months (Querol et al., 2009a). Mineral dust is then transported over the whole Mediterranean basin, following usually anticyclonic pattern of circulation (Gkikas et al., 2013) triggered by the extended subtropical anticyclone of the Atlantic Azores. Mineral dust over the Mediterranean is found to be usually transported in the lower free troposphere between 2 and 5 km of altitude (Papayannis et al., 2005; Kalivitis

et al., 2007); nevertheless, there is indication of episodes of mixing with PBL air (Bonasoni et al., 2004; Perrino et al., 2008; Pederzoli et al., 2010), with an estimated contribution to the surface particulate mass concentration of the order of 10 μg m$^{-3}$. Such studies rely on in situ measurements, while a direct evidence of mineral dust entrainment in the PBL from continuous vertically resolved observations over the Po Valley is still lacking. In this paper, to improve the understanding of the impact of such processes, we provide direct evidence of dust penetration in the lower troposphere of Po Valley, a characterization

of timing and vertical extent of its mixing with local pollution, and a direct comparison between low level intrusion and the particles concentration enhancement at the ground. Mineral dust intrusion events can indeed significantly affect public health: epidemiological studies on Italy (Sajani et al. (2010)) revealed increased respiratory mortality in correspondence of dust transport over the region. More in general, as accumulation of pollution can affect both regional climate and public health (Lelieveld et al., 2002; Kanakidou et al., , 2011), a better understanding on the processes contributing to the high concentration

of PM on the region is needed. The present paper offers more details on both local and long-range aerosol processes basing on the analyses of continuous and vertically resolved particles light scattering and depolarization in the Po Valley: in particular, Light Detection and Ranging o Laser Imaging Detection and Ranging (LiDAR) aerosol characterization, assisted by ground observations and transport models, are exploited to identify the origin of the particulate entering and mixing in the PBL during summer season. The LiDAR represents a widely used technique for studying the vertical and temporal distribution of

particulate matter optical properties (e.g., Hamonou et al., 1999; Matthias et al., 2002; Dulac and Chazette, 2013; Pappalardo et al., 2004; Amiridis et al., 2005; Papayannis et al., 2005; Flentje et al., 2010b). Coupled with in situ observations and model analysis, LiDAR observations can also be used to derive transport pathways and physical-chemical processes (e.g., Papayannis et al., 2005; Größ et al., 2013). In this work, LiDAR observations are integrated with near surface measurement techniques from observation sites at different altitudes, and coupled with Lagrangian model simulations, to provide new insights on the

processes that affect aerosol variability in the Po Valley, improving the understanding on the possible role of different local and remote sources on the PM level over the region. Please note that, from here on, by "aerosol" we will refer to just the aerosol particle phase, excluding the carrier gas.

## 2   Observations and methodology

Observations were performed in the framework of the SuperSito project, coordinated by the Regional Agency of Prevention
and Environment and funded by Region Emilia-Romagna (ARPA-ER, Italy www.supersito-er.it), and of the FP7 European project PEGASOS (Pan-European Gas-AeroSOl-climate interaction Study, pegasos.iceht.forth.gr ).



## 2.1 Measurement stations

The San Pietro Capofiume (hereafter SPC) station (44°39' N, 11°37' 0 E, 11 m a.s.l) is located in the South-Eastern part of the Po Valley, at a flat rural background site relatively close to densely populated cities and industrial sites (i.e. 30 km NE from Bologna urban area, with about 0.5 M inhabitants, and 20 km S from Ferrara, with 0.15 M inhabitants). SPC is included in the list of WMO/GAW regional stations, being representative of the surrounding wider region. The Mt. Cimone (44°12' N, 10°42' E, 2165 m) WMO/GAW global station (hereafter CMN), located 100 km south-west from SPC, is instead situated at the top of the highest peak of the Northern Apennines. For the most part of the year, CMN observations can be considered representative of the background conditions of the South Europe free troposphere (Bonasoni et al., 2000a), while, during warm months, it can be considered at a transition level between the uplifted boundary layer and the free troposphere (Andrews et al., 2011). Under summer anticyclonic conditions, polluted air masses transport from the regional boundary layer can be detected at CMN, due to thermal transport processes and PBL growth (Marinoni et al., 2008; Cristofanelli et al., 2013). CMN also represents the first mountain ridge impacted by Saharan air masses on their way across the Central Mediterranean basin to Europe.

## 2.2 Aerosol particles light backscattering coefficient profiles: LiDAR observations

The LiDAR system operating in SPC, described in Cairo et al. (2012), uses a 532 nm Nd-YAG pulsed laser source, with pulse duration of 1 ns, 400 μJ of energy and repetition rate of 1 kHz. The optical receiver of the LiDAR is a Newtonian telescope. Taking into account the distance between the telescope and the laser beam, the overlap of the laser within the FOV begins at few tens of meters (around 50 m) from the system, and is complete at around 300 m. Experimental correction allows the reconstruction of the LiDAR backscattering profile down to around 100 m, with an acceptable uncertainty (close to 10 %) on the backscatter ratio precision (see Rosati et al., 2016, and its supplementary material). Profiles are collected every 10 minutes with a vertical resolution of 7.5 m extending, on average, up to 7 km. In the following discussion, we will make use of backscattering ratio ($R$) and aerosol depolarization ($\delta a$) , defined as (Browell et al., 1990; Cairo et al., 1999):

$$R(r) = \frac{\beta_a(r) + \beta_m(r)}{\beta_m(r)}$$

$$\delta_a = \frac{\beta_{a_{perp}}}{\beta_{a_{par}}}$$

$\beta_m(r)$ represents the backscatter coefficient from molecular contribution while $\beta_{a_{perp}}$ and $\beta_{a_{par}}$ represent the backscattered signal components from aerosol light scattering, with polarization respectively perpendicular and parallel to the polarization of the emitted light. The inversion of the LiDAR signal is accomplished with the Klett method (Klett, 1985; Fernald , 1984), finding a suitable region of the profile that is supposed to be free of aerosol to calibrate the signal, and using piecewise constant extinction-to backscattering ratio (LiDAR ratio, $L$) values. We make use of different values for $L$ following what reported in literature: desert dust (identified by $\delta_a(r) > 10\%$) is characterized by $L$ equal to 50 sr (Müller et al., 2007) while for low depolarizing aerosol we assume the values typical for anthropogenic aerosol, L~60–70 sr (Murayama et al., 1999; Ferrare et al., 2001; Fiebig et al., 2002). In addition we considered different values for water cloud ($L = 20$ sr) and ice clouds ($L = 30$ sr)





(Chen et al., 2002; O'Connor et al., 2004). A more detailed description of the methods used to perform the inversion of LiDAR data, together with a through uncertainty analysis, performance in conditions close to the SPC site and additional experimental set-up details, are given in Cairo et al. (2012) and Rosati et al. (2016) and its supplementary material.

### 2.3 Aerosol particles number size distribution: APSS and OPSS

Aerosol concentrations at the ground are obtained from an Aerodynamic Particle Sizer Spectrometer (Type TSI, APS model 3321), operating at SPC, that provides real-time aerodynamic measurements of particles from 0.5 to 10 μm at 1 minute time resolution. The aerodynamic diameter is defined as the physical diameter that a unit density sphere will have if settles through the air with a velocity equal to the one of the sampled particle. The aerodynamic diameters of particles is established measuring their transit time between two points when accelerated singly through a well-defined flow field. The aerodynamic diameters are

here converted to volume-equivalent particle diameters, following Khlystov et al. (2004) and assuming an effective particles density of 1.8, during dust days, and 1.55 during the remaining days, accordingly to the values retrieved by Putaud et al. (2004). The near surface aerosol number concentrations at the free tropospheric level are instead derived from an Optical Particle Size Spectrometer (Type Grimm, OPSS Particle Size Analyser Mod. 1.108) operating at the CMN station. The OPSS provides particle counts in the diameters ($Dp$) range of $0.3\mu m < Dp < 10\mu m$. The instrument is based on the quantification of the $90°$

scattering of light by aerosol. According to the specifications, the reproducibility of the OPSS in particle counts is $\pm 2\%$ (Putaud et al., 2004). Such measurements allow the determination of the fine ($0.3\mu m < Dp < 1\mu m$) and coarse ($1\mu m < Dp < 10\mu m$) aerosol fractions with a 1 minute time resolution. For the purpose of the paper we make use of the time series of coarse ($Dp > 1\mu m$) aerosol concentration observed at CMN, without any further correction to the "optical" diameter, to provide a clear indication on the presence of mineral dust layer in the free troposphere (Marinoni et al., 2008; Duchi et al., 2016).

### 2.4 Chemical Composition: MARGA

The Monitor for AeRosol and GAses (MARGA, Metrohm Applikon B.V. Schiedam NL) is a wet chemistry instrument that provides continuous measurements of the water soluble inorganic gases and aerosol components that might have a direct effect on air quality (Makkonen et al., 2012; Rumsey et al., 2014). The analytical system allows for the characterization of inorganic aerosol ($Cl^-, NO_3^-, SO_4^{2-}, NH_4^+, K^+, Ca^{2+}, Mg^{2+}, Na^+$) and gases ($NH_3, HNO_3, SO_2, HONO, HCl$) at hourly

resolution (Twigg et al. , 2015). In the sampling box, the air passes through a Wet Rotating Denuder (WRD) (Keuken et al., 1988) where water soluble gases are stripped from the air stream and collected in water. The sampled air then continues through a Steam-Jet Aerosol Collector (SJAC, Khlystov et al., 1995; Slanina et al., 2001) where the water soluble aerosols are separated from the air stream and collected. The gas and aerosol samples are then analysed by online ion chromatography with high accuracies (detection limits as low as 0.01 μg m$^{-3}$ (Twigg et al. , 2015)). Size-selective particle cyclones are used in front

of the two MARGA sampling boxes so that the size of the particles for analysis can be limited to an aerodynamic diameter of less than 10 (PM10) or 1 (PM1) μm.





## 2.5 Transport modelling

We make use of the FLEXPART Lagrangian particle dispersion model (version 9.02) to characterize the transport during summer 2012 (Stohl et al., 2005, and references therein). FLEXPART is a widely used model system to simulate synoptic and mesoscale transport and diffusion of aerosol and trace gases, as well as loss processes such as dry and wet deposition or ra-
dioactive decay (Stohl et al., 2005), and has been validated using large-scale tracer experiments (Stohl et al., 1998; Forster et al., 2007). In our case, the model is driven by pressure level data from NCEP Global Forecast System (GFS) (rda.ucar.edu). Meteorological input is provided each 6 hours (00:00, 06:00, 12:00 and 18:00 UTC) at a resolution of $0.5° \times 0.5°$. To assess the effective role of Saharan dust transport on aerosol concentration over Northern Italy, 5-days backward plumes are coupled with mineral dust emissions from Africa taken from the DREAM inventory (Pérez et al., 2006a, b; Basart et al., 2012, web-
site http://www.bsc.es/earth-sciences/mineral-dust-forecast-system/bsc-dream8b-forecast) at $0.2° \times 0.2°$ horizontal resolution. FLEXPART gives as output the footprint of the retroplume, namely the time of residence of the air parcels over each geographical grid point, during the 5 days prior to the moment of the trajectories release. Such quantity, expressed in seconds, gives an indication of which, and in which extent, emissive regions are going to contribute to the mineral dust enrichment of the air parcels, influencing therefore dust burden at the time and position of trajectories release. The product of the footprint with
the dust emission rate (given in $\mathrm{kg\,m^{-2}h^{-1}}$) is then integrated over the whole geographical domain to give an estimate of the mass of mineral dust advected over SPC for each back-plume release. For simplicity, dust on emissive areas is considered to be injected uniformly below $1000\,\mathrm{m}$ a.g.l., therefore only trajectories crossing this height are included in the footprint-emissions coupling. The backtrajectories clusters were released every 6 hours, along the whole campaign period, from the SPC station in correspondence of the $1000 - 2000\,\mathrm{m}$ and $3000 - 4000\,\mathrm{m}$ atmospheric layers.

## 3 LiDAR aerosol type classification

LiDAR observations have been extensively used to identify mineral dust layers and discriminate among different typologies of aerosols, based on a choice of specific ranges of optical parameters considered representatives of distinct aerosol types. Examples are shown in Burton et al. (2012), where the classification among eight different types of aerosol is derived from total depolarization ratio ($\delta$), LiDAR ratio ($LR$) and the color ratio ($CR$). In Größ et al. (2013) the categorized aerosol types
are: sea-salt, mineral dust and mixed dust based on $LR$ and aerosol depolarization $\delta_a$. The estimate of $LR$ (that requires an independent information on the extinction that should be derived from the Raman signal), as well as an evaluation of the $CR$ (based on the adoption of two wavelength channel), is not possible with single wavelength elastic LiDAR as the one deployed at SPC during the PEGASOS campaign. Nevertheless, some typologies of aerosol show distinct values of aerosol depolarization. At $532\,\mathrm{nm}$, values of aerosol depolarization around or higher than $30\,\%$ are generally associated with layers of
nearly pure mineral dust while smaller values (around 8–10 %) are often detected in correspondence of mixture of mineral dust and non-depolarizing particles (Murayama et al., 2003; Sugimoto et al., 2006; Tesche et al., 2009a). By contrast, smoke and other anthropogenic aerosols exhibit low values of aerosol depolarization (less than 5 %)(Sun et al., 2012; Größ et al., 2013).



Following the results of the abovementioned studies, here we implement a three-types aerosol discrimination scheme to characterize the vertical and temporal aerosol variability over the region along the campaign period (15 June 2012 – 5 July 2012). The reader should notice that the lower depolarization values that we observe respect what usually found in literature (especially for the dust layers) are more likely linked to the calibration process, and in particular to the difficulty in individ-uating completely aerosol free layers in the vertical span of the adopted LiDAR system (from ground to 7 Km). The LiDAR classification, based on the statistical distribution of the overall observed $\delta_a$ and $R$ values, is also applied here to overcome such limitations. The robustness of the results are then further supported by comparison with lagrangian analysis and in-situ measurements.

Figure 1 reports the probability density function, along the whole measurements campaign, of the aerosol occurrence, ex-pressed as a function of $1 - 1/R$ (ranging from 0, in aerosol free condition, to 1 when $R$ tends to infinity) and $\delta_a$. The analysis includes 10 minutes resolution observations from the ground up to $5000 \, m$ height, for a total of about $1.5 \cdot 10^7$ sampling points. Aerosol, identified with values of $1 - 1/R$ larger than 0.2, can be further discerned in three distinct patterns:

1. low values of $\delta_a$ (< 3 %); based on the above references, these particles may be composed of anthropogenic pollution and, for higher values of $R$, by droplets, and are defined as non-depolarizing.

2. high values of $\delta_a$ (> 10 %); according to the previously mentioned literature, this can be consider as a threshold value for mineral dust or mixed dust particles. This class is defined as depolarizing.

3. intermediate $\delta_a$ values (3 %< $\delta_a$ <10 %) which, based solely on $R$ and $\delta_a$, cannot be considered as indicative of a dominance of a defined aerosol type unless coupled to a more thorough correlation with additional observations. We will refer to this type as intermediate depolarizing.

The boxes in Fig. 1 report the $R$ and $\delta_a$ ranges for each of the three classes used to derive the aerosol mask for the whole campaign. The results are reported in Fig. 2 together with the profiles of scattering ratio $R$ and aerosol depolarization $\delta_a$. Overall, non-depolarizing particles (type 1) are dominant throughout the campaign with a total occurrence of 49 % of the measurements in the 100-5000 $m$ range. They are observed prevalently below 2000 $m$ height, and are associated with enhanced values of $R$ (parameter $1 - 1/R$ ranging between 0.3 and 0.6). Due to the presence of such particles, the vertical gradient of $R$ marks, when not masked by the presence of mineral dust layers or clouds, the PBL evolution. Two events of depolarizing aerosol (19 June – 21 June and 29 June – 02 July), recognized as type 2 and likely related to mineral dust presence, are observed between 2000 $m$ and 5000 $m$. Such events are also clearly visible as enhancement of $R$ in the free troposphere with values of $1 - 1/R$ ranging from 0.6 and 0.8. The intermediate class (detected with an occurrence of around 19 % during the whole campaign) is found in close proximity of the depolarizing aerosol and within the PBL. The observed vertical and time distribution of these intermediate type particles indicates the possibility of mixing of the dust depolarizing layers with local non-depolarizing particulate. Nevertheless, intermediate $\delta_a$ values are also observed systematically after 12:00 UTC (Universal Time Coordinate) between 0 and 1500 $m$ height for the majority of dust-free days. The nature of such intermediate non-dust depolarizing aerosol is further discussed in Sect. 6.





## 4  Meteorology and synoptic aerosol regimes

The evolution of the meteorological conditions at SPC is reported in Fig. 3. Vertical profiles from ground up to 4 km height of wind speed and direction come from radiosondes (Vaisala RS92) launched daily at 00:00, 06:00 and 12:00 UTC. Figure 3 reports also ground temperature at 12:00 UTC. PBL evolution description is referred to the PBL height time series presented in Sandrini et al. (2016) and reported also in the supplementary material (see Fig. S1). The observation of wind profiles over SPC highlights a sequence of distinct meteorological regimes:

- Stagnation period: (15 June – 19 June) This first phase is characterised by a situation of stagnant conditions (wind speed less than 4 m s$^{-1}$ below 2000 m height) with a progressive warming of the air masses (from 29 °C up to 34 °C at the ground). The PBL top is limited below 1500 m until 19th June when it reached 2000 m.

- South-westerly winds: (20 June – 21 June) During this phase higher wind speeds (between 11 m s$^{-1}$ and 16 m s$^{-1}$) are observed above 2000 m. Correspondingly, wind direction profiles indicate a prevalent South and South-West provenience. The arrival of warm Saharan air masses (temperature at the ground around 32°C–33°C) led to a more intense PBL development (up to 2000m) respect to the previous stagnation phase. (30 June – 5 July) During the last days of the campaign, strong winds (between 12 and 15 m s$^{-1}$ with a peak of 20 m s$^{-1}$ on 3 July) are observed above 1500 m. While during the 30 June winds are coming mainly from South, the following days are characterized by a steering to south westerly (1 July) and then westerly flow (2 and 3 July). During this phase, temperatures at the ground reached 34°C–35°C but the dust layer presence made it difficult to unambiguosly retrieve the PBL top. During the immediatly following days (3–4 July), the PBL top was detected to be above 2000m.

- North-easterly winds: (22 June – 29 June) The radiosounding profiles indicate a prevalence of south-easterly or easterly winds above 2000 m a.g.l.. Northerly/north-easterly winds are instead often visible at lower altitudes, in particular between 23 and 24 June below 1000 m, on 26 June below 500 m (associated also to wind intensities up to 10 m s$^{-1}$) and on 27 June between 500 m and 1800 m. During this period, ground temperatures first decrease to 30 °C then increase again after 27 June, up to 33 °C. The PBL maximum height varies between 1500 m and 2000 m a.g.l.. Such conditions are favourable for the export of the Po Valley pollution toward the Tyrrhenian Sea and will be extensively discussed in a companion paper.

The evolution of the observed size distribution and optical classification during the distinct meteorological regimes is presented in the following sections.

### 4.1  Summer stagnant conditions: 15–19 June

Meteorological evolution is integrated with the aerosol optical variability from LiDAR (see Fig. 2) and with ground aerosol number concentration and volume size distribution (estimated as the volume of a sphere of diameter corresponding to the volume-equivalent particle diameter) at SPC and CMN (see Fig. 4). Panel a of Fig. 4 shows the time trend of small particles concentration (297 nm $< Dp <$ 420 nm) from the APSS at SPC. The stagnation period (15–19 June), typical for the Po Valley on





hot summer days (Rossa et al., 2012), is characterized by a marked daily cycle in the aerosol concentration and by a progressive day-by-day accumulation of particles in the PBL. This is noticeable in the increase of the particle number size distribution of small particles from APSS (from 5 $cm^{-3}$ to nearly 20 $cm^{-3}$) during the early morning hours (00:00-06:00 UTC), when the lower troposphere is stably stratified. The LiDAR observations (Fig. 2) show a persistent layer of non-depolarizing particles up

to 2 km height, attributable to anthropogenic aerosol and modulated vertically by the PBL daily cycle. APSS data in panel b of Fig. 4 show a bimodal aerosol distribution with a clear increase of volume mode due to submicron particles (0.5 μm < $Dp$ <1 μm) growing from 0.4 $μm^3cm^{-3}$ to more than 1 $μm^3cm^{-3}$. During this period of the campaign, the OPSS at CMN (panel c Fig. 4) indicates coarse particle number concentrations below 0.4 $cm^{-3}$.

### 4.2   Saharan dust events: 19–21 June and 28 June–3 July

During the first event (19 to 21 June) strong south-westerly winds (with speeds greater than 10 $m\ s^{-1}$ above 2000 m height) are associated with a stable anticyclonic circulation centred above South Mediterranean and Tunisia, leading to efficient south-south westerly circulation. Mineral dust can be clearly observed as an enhancement in LiDAR $R$ profiles (Fig. 2, panel b) above 2000 m until June 21 while, below that height, it is not possible to discern any deviation from the background aerosol signal. The enhancement in $R$ is accompanied with increased aerosol depolarization ($\delta_a \sim 10\% - 15\%$) during the whole

event, with values up to 20 % above 3000 m during the 20 June, resulting in a coherent layer of type 2 particles visible in the aerosol mask (panel a of Fig. 2). In correspondence of the presence of dust aerosol at 2000 m level, the OPSS at CMN (panel c, Fig. 4) detects an increase of coarse particles concentration up to 1.8 $cm^{-3}$. The peak seen on 21 June at around 09:00 UTC (greater than 5 $particles\ cm^{-3}$) should be attributed to an enhancement in aerosol load, that can be both caused by an intensification of mineral dust burden or, as suggested by a corresponding increase in black carbon concentration observed

at CMN (see also Cristofanelli et al. (2016)), by mixing with pollution from the regional PBL (Cristofanelli et al., 2009). Intermediate depolarization aerosol type 3 is observed below the depolarizing layer, throughout the mineral dust event, and persisting until 22–23 June. As mentioned in the previous section, this can be a signature of mineral dust mixing with local non-depolarizing particulate. The comparison of the aerosol layer structure with the in-situ measurement indicates, in fact, a simultaneous enhancement of coarse (2 μm < $Dp$ <5 μm) particles observed by the APSS on 20-22 June (panel b of Fig. 4 )

in correspondence of the type 3 class near the ground, suggesting mineral dust presence in the layer composition. It's worth to notice that, while ground measurements do not indicate a clear coarse particle enhancement after 22 June, the LiDAR still observed a lofted layer of intermediate depolarizing aerosol until 23 June. During the second event (28 June to 3 July) high wind speeds above 2000 m (up to 20 $m\ s^{-1}$) are associated with a high pressure area centred above central Italy, leading again to favourable south-westerly circulation. LiDAR data show a second layer of enhanced $R$ ($1 - 1/R$ around 0.6) lasting from

28 June at 23:00 UTC to July 3 at 00:00 UTC. Depolarization reaches values higher than in the previous event (with mean values of 15 % and maximum exceeding 20 %); this is again visible as a thick and persistent layer of type 2 aerosol that, in this case, extends down to the ground on 1 July. As in the previous case, it is possible to observe the presence of intermediate depolarization particles (type 3) close to the depolarizing layer (type 2). The dust layer appears characterized by a more intense contribution of coarse particles respect to the previous event, visible both at lofted level and ground. In correspondence of the





presence of a depolarizing layer at 2000 m, the OPC (panel c, Fig. 4) shows coarse particle concentrations nearly doubled compared to the previous event (concentration between 2 and 3 particles $\mathrm{cm}^{-3}$). Similarly, in situ observations at SPC (panel b, Fig. 4) show an increase in coarse particles volume simultaneously to detection of type 2 and 3 particles close to the ground, with values higher than in the previous event ($>1\,\mu\mathrm{m}^3\mathrm{cm}^{-3}$) and increased contribution from the intermediate particles sizes

($1\,\mu\mathrm{m} - 2\,\mu\mathrm{m}$).

The upper panels of Fig. 5 shows the footprints of the 5 days retro-plumes released on the 20 June at 18:00 UTC (panel a) and on the 29 June at 12:00 UTC (panel b). The transport for the first event has a more direct pathway, with an average transport time of 2 days from North Sahara. The second event appears instead to originate from Western Sahara and has a longer pathway revolving around the anticyclonic circulation (around 4 days). Aerosol Optical Depth from multi-model forecasts

(SDS-WAS Sand and Dust Storm WMO warning advisory and assessment system, visible at http://sds-was.aemet.es/forecast-products/dust-forecasts/compared-dust-forecasts) indicates a spatial distribution in agreement with the FLEXPART footprints for the two events. The simulated mineral dust concentration over the SPC site, derived by the coupling of FLEXPART footprint with DREAM emissions (panel c, Fig. 5), allows to provide an estimate of the evolution (with time step of 6 hours) of the mineral dust concentration on the SPC site. The simulation confirms the presence of the two desert dust transport periods and

the progressive descent of the dust layers advection from 3000–4000 m to 1000–2000 m heights, simultaneously to what shown by LiDAR observations. Maximum mineral aerosol load from FLEXPART analysis occurs on June 20, both at the upper layer ($12$–$13\,\mu\mathrm{g\,m}^{-3}$) and at the bottom layer ($11\,\mu\mathrm{g\,m}^{-3}$). These concentrations lie in the range of past observations collected over the Northern Mediterranean area (estimated to be around $10\,\mu\mathrm{g\,m}^{-3}$ during June–July months)(Pey et al., 2013). According to FLEXPART, the import of mineral dust persists until the morning of 23 June, when dust presence is not unambiguosly inferable

from observations but the aerosol mask still indicates the presence of intermediate depolarizing particles below 2000 m. The second desert dust event predicted by FLEXPART shows the same timing with respect to observations but, while the APSS and the OPSS indicate a similar dust burden for the two desert aerosol advection events, the dust load indicated by the model (between 3 and $5\,\mu\mathrm{g\,m}^{-3}$) appears lower respect to the previous events. It should be emphasized that a quantitative assessment of the mineral aerosol transport is difficult, due to uncertainties related to the fixed height of PBL over desert for mineral dust

uptake and to uncertainties on the emission estimate and on trajectories dynamics (Stohl et al., 1998). Despite such limitations, the model offers a robust characterization of the dynamics and timing of the events, supporting the interpretation of the data analysis.

## 5   Mean daily variability

Figure 6 reports the frequency of observations for each of the three classes, integrated for the period 15 June – 5 July. Depolar-

izing aerosol (panel a, Fig. 6) are associated with the two events of desert dust and hence are mostly observed between 1500 and 5000 m height with a frequency of occurrence ranging between 15 % and 30 %. Non-negligible occurrences ($\sim 10\,\%$) are also observed close to the ground and can be attributed to the mineral dust descent during the second event. Non-depolarizing aerosol (panel c, Fig. 6) is dominant throughout the campaign (occurrence up to 80 % below 2000 m); This class of aerosol




appears to be mainly confined below the PBL top (derived by the analysis shown in Fig. S1 of the supplementary material and traced with a black dashed line). During the campaign, diurnal PBL starts to develop on average at 06:00 UTC and reaches its maximum vertical extension, up to 2 km height, between 17:00 and 18:00 UTC. An high occurrence of non-depolarizing particles marks the PBL average daily evolution both during the diurnal formation and at night-time (21:00 to 05:00 UTC), forming

the residual layer. A clear minimum of non-depolarizing particles occurrence is observed in the afternoon between 16:00 and 19:00 UTC, when LiDAR indicates instead a maximum (50 % of observation) of intermediate depolarization type occurrence (panel b, Fig. 6). Such enhancement of type 3 particles detection during late afternoon appears frequently along the campaign (clearly visible on 13 days over 21, see Fig. 2). It should be noticed that on 20–22 June and 30 June–2 July the presence of mineral dust can mask any $\delta_a$ enhancement in the PBL. On average the vertical extent of such layer of intermediate depolarization

is limited within the PBL below 1500 m height (panel b, Fig. 6). The mean diurnal evolution of the different classes, integrated temporally between the four synoptic hours (00:00 UTC, 06:00 UTC, 12:00 UTC, 18:00 UTC) and vertically, below and above 2000 m, is summarized in Fig. 7. Overall, aerosol are mainly observed below 2000 m with non-depolarizing aerosol being the predominant type (between 50 % and 60 % of the observations through the whole day). Intermediate depolarizing type shows its diurnal cycle with an increase in detection from 9 % in the early hours to 22 % during afternoon and late evening.

Desert dust is observed at ground with a non negligible occurrence (7 %) during the campaign. Aerosol is observed above 2000 m in less than 50 % of the observations, mainly associated to mineral dust presence (contributing from 13 % to 21 % of the detections). Average daily particle volume size distribution evolution, relatively to dust free days, is reported in Fig. 8. Fine particles ($Dp < 1\ \mu m$) volume shows a semidiurnal cycle, corresponding to the diurnal cycle of non-depolarizing aerosol near the ground, with concentration increasing during the stable nocturnal layer phase (late night - early morning) and strongly

decreasing during the stage of well-developed PBL. The larger particles mode shows two maxima: a first one (volumes < 0.4 $\mu m^3\ cm^{-3}$), in correspondence to the uplift of the PBL layer around 9:00 UTC, and a second one forming at 15:00 UTC, with a maximum (volumes > 0.5 $\mu m^3\ cm^{-3}$) at 20:00 UTC, showing a similar timing than the depolarization enhancement described above. Further analysis on the afternoon PBL aerosol composition is reported in the following section.

## 6   Non desert dust depolarising aerosol

We report in Fig. 9 the $\delta_a$ profiles of a representative case study (3 July) of the late afternoon occurrence of intermediate depolarizing aerosol. The aerosol depolarization indicates that the plume starts to develop from 15:00 UTC to 20:00 UTC and reaches the maximum height of 1500 m in late evening, with a vertical structure suggesting a possible uplift of particles from the ground. An increase in LiDAR aerosol depolarization in regime of convective PBL was already observed by Gibert et al. (2007a). Their results show a positive correlation of enhanced $\delta_a$ with an increase in vertical wind velocity, possibly indicating

a source emission of particles transported upward by convection. The actual nature of the aerosol plume cannot be assessed solely by LiDAR depolarization. An increase in depolarization can be due to the presence of irregularly shaped particles that can belong to a wide range of aerosol type, from soil and desert dust, to marine aerosol (Murayama et al., 1999) and ash particles (Nisantzi et al., 2014). The hourly time resolution measurements of PM1 and PM10 aerosol chemical compositions,





provided by the MARGA analyser, show no evident correlation between the depolarization increase and the presence of sea salt (not reported here). Similarly, no correlation was found with absorbing aerosol (black carbon), investigated by means of a multi-angle absorption photometer (MAAP, Petzold et al., 2006b) (also not shown). By contrast, MARGA observations highlight a maximum in PM10 calcium concentrations, simultaneously to the afternoon increase in ground depolarization

(starting between 15:00 and 20:00 UTC, see panel b of Fig. 6) and in the larger particles detection from APSS (maxima between 18:00 and 20:00 UTC, Fig. 8). The diurnal mean evolution of the calcium ion (Ca2+) fraction, calculated over the total PM10 ions, shows a marked increase after 10:00 UTC with a maximum in the late afternoon (17:00 UTC – 20:00 UTC, up to 0.35, see panel a , Fig.10). This daily behavior is in agreement with the enhancement in aerosol volume contribution from large particles respect to the fine ones, shown in panel b of Fig. 10. These results reinforce the hypotesis of the crustal origin

of the intermediate depolarizing particles observed by the LiDAR. It is possible therefore to explain, at least on qualitative basis, the recurrent detection of the afternoon aerosol plumes as emissions and resuspension of soil particles from dried land sources. The frequent occurrence of such events during the observational campaign indicates that the Po Valley can effectively act as a source of mineral particles, likely originated from agricultural soils, that under convective atmospheric conditions can be uplifted at the PBL top in late afternoon hours. This is further confirmed by the diurnal evolution of non-desert dust

coarse particles concentration at CMN (panel c, Fig. 10), that indicates an enhancement in the coarse particles fraction in late afternoon/evening. Hence, recurrent vertical transport from the Po Valley, triggered by thermal air-mass, can uplift mineral aerosol over the mountains ridge, potentially impact on particulate transport up to the regional scale (Cristofanelli et al., 2016).

## 7 Effect of aerosol hygroscopic growth on aerosol particles light scattering and depolarization

LiDAR data (Fig. 2) frequently show, during early morning hours, a shallow layer of non-depolarizing aerosol below 300

m height, more easily visible during days characterized by desert dust and mixed dust events (see for instance 00:00-06:00 UTC of the 19 June and between 00:00-08:00 UTC of 30 June). The decrease in depolarization is less evident in dust-free atmosphere but is nevertheless observed in several other days of the campaign (18 June, 21 June, 22 June, 29 June, 1 July, 4 July and 5 July), always below 300 m, before 08:00 UTC and usually associated to high values of $R$ $(1 - 1/R > 0.6)$. A detailed time-height evolution of $\delta_a$ and relative humidity ($RH$) for the 30 June (here considered representative of the abovementioned

events) is reported in Fig. 11. The mineral dust plume is clearly visible throughout the depolarization profile. Between 1000 m and 4000 m height, higher depolarization ($\delta_a > 10$ %) is associated to dry air ($RH < 50$ %). Below 1000 m height, RH varies between 50 % and 70 % and $\delta_a$ shows lower values compatible with dust presence mixed with local pollution (around 4 %). Conversely, the lowermost troposphere (below 300 m) is characterized by a layer with a sharp decrease in $\delta_a$ (less than 2 %) associated to a marked increase in $RH$ (larger than 80 %). The study, extended separatedly on the whole dust events

period (20–23 June and 29 June – 2 July, lower panel of Fig. S2, supplementary material) and in the remaining dust free days (upper panel of Fig S2, supplementary material), indicates in both case a depolarization decrease in the lower layers, visible starting from RH>60%. During dust free days the affected aerosol layer (below 400 m) shows a depolarization decrease of about 1% (from an average value of 2% to values around 1% and less). During dust days the process influences the aerosol





layers at different levels: for RH values between 60% and 65%, depolarization values decrease is around 2% (from more than 7% to around 5.5%). With increasing RH (RH>70%) the effect is more evident, with a decrease down to 3.5%. This low depolarization near ground suggests the presence of increasingly spherical particles, which can be originated by two different processes:

– the presence of fine particles of anthropogenic origin that may deliquesce: The stagnant meteorological conditions that characterize the Po Valley during anticyclonic phases are favorable for the formation of secondary inorganic aerosols (especially ammonium nitrate) and of secondary organic aerosol (Sandrini et al. , 2016). A recent study (Hodas et al., 2014) showed that, during the same 2012 campaign at SPC, the aerosol liquid water was mainly driven by locally formed nitrate, hence growth of spherical non-depolarizing aerosol could occur due to deliquescence of fine particles of
anthropogenic origin, of which nitrates were the dominant compound.

   – Mechanisms explaining the increase of scattering of mineral dust particles, along with a reduction of their depolarization ratio, can also be hypothesized (Ikegami et al., 1993; Murayama et al., 1999; Sassen et al., 2002; Zhou et al., 2002; Nee et al., 2007). It should be emphasized that, during the analysed case study, high relative humidity values (80 % or more) are observed in the lowermost non-depolarizing layer (see Fig. 11), suggesting that condensation of water around mineral
dust particles coated by (or simply enriched of) hydrophilic components may play a role in the modification of the optical properties of desert dust in this atmospheric layer. Indeed, even if mineral dust is primarily a hydrophobic aerosol, it can become hydrophilic due to chemical reactions occurring on the particles surfaces during long-range transport (Nee et al., 2007; Sullivan et al., 2009a) or locally from condensation of inorganic and organic soluble materials from ground sources.

During the summer 2012 campaign, under the observed conditions, both processes may have played a relevant role: Fig. 12 shows that, during the stagnation phase (from 14 June until 19 June), aerosol nitrate (NO3-) concentrations in both fine and large particles MARGA channels (PM1 and PM10) increase with a marked daily cycle peaking at night. The submicron fraction of nitrate dominates the concentration of PM10 nitrate during this phase. The APSS submicron particle volume concentration follows a daily variability and a buildup similar to what shown by the APSS nitrates concentrations, reaching maxima during
the 19 June (APSS volume concentration up to 25 $\mu$m m$^{-3}$ and NO3- PM1 and PM10 concentration up to 15 $\mu$g m$^{-3}$ and 18 $\mu$g m$^{-3}$, respectively). Such increase, evident during early morning hours in small aerosol from APSS and in PM1 nitrate from MARGA, supports the hypothesis of deliquescence of anthropogenic fine particles explaining the low aerosol depolarization observed by the LiDAR in the surface layer, particularly during dust-free days. Similarly, under dust presence, the low depolarization values can be related to external mixing of dust depolarizing particles with such locally formed spherical
particles. Nevertheless, after the end of the stagnation phase (19 June) the aerosol nitrate concentration decreased and, during the observed desert dust episodes, the difference between the nitrate PM10 and PM1 fractions became more evident, with PM10 prevailing over PM1. The intensified ventilation established after 19 June may in fact have limited the accumulation of anthropogenic particles, at the same time carrying dryer African air masses and making nitric acid condensation on coarse





particles prevail over condensation on accumulation mode aerosol. Consequently coarse-mode nitrate would promote water condensation on the large particles, leading to low aerosol depolarization values even in presence of dust.

## 8  Conclusions

The presented paper provided a characterization of the effects of meteorological evolution and transport patterns on the aerosol variability, based on the observations collected during two major field campaigns (PEGASOS and SuperSito) in the eastern part of the Po Valley. The aim was to contribute to the understanding of the processes that lead to the high concentration and variability of aerosol characterizing the Po Valley. The central focus was studying the evolution of the aerosol depolarization profiles during typical summer conditions. In particular we were able to describe the evolution and duration of dust intrusions in the PBL. In addition we observed two other processes (late afternoon particles resuspension from the soil and early morning hygroscopic growth), reasonably related to regional activities (i.e. farming or industrial and combustion processes), that can impact on PM concentration in the PBL. We offered therefore a qualitative explanation and understanding of such processes. The analysis of meteorological conditions, coupled with observations from LiDAR and in situ aerosol number/size distribution spectrometers, led to the identification of distinct meteorological regimes with a temporal and spatial distribution of different aerosol types: a first phase (15–18 June) was characterized by a stagnation period (weak winds below 2000 m), representative of hot and polluted conditions in the Po Valley whole area, with progressive accumulation of locally emitted aerosol in the lower troposphere and consequent increase of the fine mode aerosol concentration near the ground. Particles concentration at the ground showed therefore a clear diurnal cycle, reaching maxima during the early morning, when PBL uplift and vertical mixing were absent. Observations and Lagrangian analysis allowed a detailed description of two events of Saharan dust transport (in line with the average occurrence of 2–3 summer desert dust episodes over the region detected by satellite (Gkikas et al., 2013)). Mineral dust layers were advected over the measurement site from the Sahara desert, traveling along anticyclonic patterns at high level (around 3000–4000 m) and carrying depolarizing aerosol. The in-depth analysis of such events offered also evidence of vertical mixing of desert dust with local pollution. In both the events the plumes descended indeed to low height (with a total occurrence of depolarizing aerosol identification inside the PBL of ∼7 % along the whole campaign), leading to the detection of coarse particles (Dp > 1 μm) at the ground. As, on climatological basis, Saharan dust advection occurs with noticeable frequency over Northern Mediterranean (i.e. Pey et al. (2013) indicated a frequency of 17 % for the 2001–2011 period), dust intrusion can represent a significant factor in increasing PM concentration at the ground. Such results give direct evidence to the suggestion of Bonasoni et al. (2004) that hypothesized, basing on in situ measurements in North Italy and back-trajectories analysis, that mineral dust events detected in free troposphere can possibly lead, with non-negligible frequency, to PM10 exceedance at ground in the time span of some hours.

Less known processes, as recurrent uplifts of soil aerosol and a likely hygroscopic growth of particles in presence of mineral dust, were also frequently observed during the different phases of the campaign. The existence of a contribution to PM10 levels from resuspension aerosol sources in European regions was already hypothesized by Vautard et al. (2005), based on chemical transport model study. Here, several events of intermediate depolarizing aerosol (mean diurnal frequency of detection





~22 %), up to 2000 m height, were observed during late afternoon (17:00–20:00) in non-dust days. The concurrent increase in calcium particle spectroscopic measurements (with contribution up to 0.35 of the total PM10 fraction) indicated the crustal nature of such aerosol, and can therefore be reasonably attributed to processes of vertical uplift of soil particles. The vertical extension of such plume, as observed in LiDAR profiles and in the diurnal variability in the CMN measurements, suggests

also that local pollution can be transported above mountains peak and therefore potentially exported outside the orographic boundaries of the region. The combination of depolarization profiles with meteorological and aerosol measurements allowed also to identify the presence of a shallow layer of low depolarizing ($\delta_a < 2$ %) aerosol at the ground, in correspondence to high relative humidity values ($RH > 80$ %). The timing and the high values of nitrates ions concentration (up to 18 μg m$^{-3}$) in the PM1 and PM10 channels, indicated that the origin of such low depolarization particles can be related to processes of secondary

organic aerosol formation and hygroscopic growth on mineral dust particles with nitrate-enriched surface. In conclusion, the in-depth analysis of the aerosol light backscattering profiles provided new insights on particles behaviour from ground up to free troposphere. Results pointed out particles processes, observed relatively frequently on the time span of the campaign, impacting aerosol variability, air quality and, potentially, regional climate and deserving therefore more extended analysis from longer-period vertical resolved observations (i.e. EARLINET network). The detailed retrieved information (vertical stratification,

hygroscopic growth near ground, aerosol evolution inside the PBL) can also serve as a support for larger scale studies. We cite here as an example a recent study, based on MAIAC satellite information (Arvani et al. , 2016), that tries to assess a method for surface PM retrieval from space observations relying on rough approximations of PBL evolution and RH effect on aerosols. Accurate studies on such aspects, as the one proposed here, can therefore represent an important contribution in the improvement of more complex and focused atmospheric observation techniques.

*Competing interests.*  The authors declare that they have no conflict of interest.

*Acknowledgements.*  This work was partly funded by the projects PEGASOS (FP7-ENV-2010-265148), the project SuperSito by the Emilia-Romagna region(DRG no. 428/10), the EU project StratoClim (grant agreement no. 603557), the EU FP7 grants ÉCLAIRE (grant 282910) and the project of National Interest NextData. We would like to acknowledge the Energy Research Centre for the Netherlands (ECN) to provide us with a MARGA instrument to use during these campaigns. This study received fundings also from the FP7 project BACCHUS

(grant agreement 603445). Dust emission used in this work come from the BSC-DREAM8b (Dust REgional Atmospheric Model) model, operated by the Barcelona Supercomputing Center http://www.bsc.es/projects/earthscience/BSC-DREAM/).





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



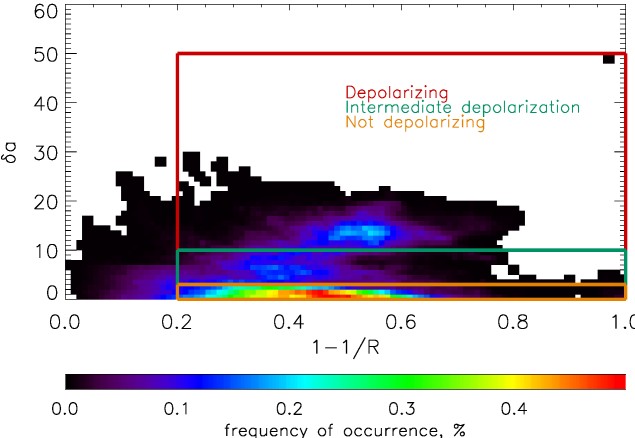

**Figure 1.** Probability Density Function of aerosol particles optical properties over the 15 June – 5 July 2012 as a function of $1 - 1/R$ and $\delta_a$ parameters. The color code indicates the frequency of occurrence





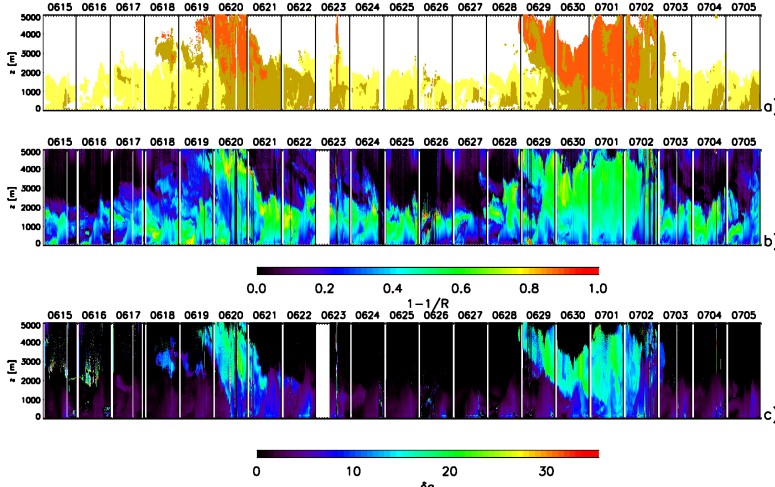

**Figure 2.** LiDAR observations along the whole campaign. Panel a: vertical profiles of aerosol particles types resulting from the classification methodology described by Sec. 3: not depolarizing aerosol (yellow), depolarizing aerosol (orange) and intermediate aerosol (brown) properties. Panel b: vertical profiles of $1 - 1/R$. Panel c: vertical profiles of $\delta_a$



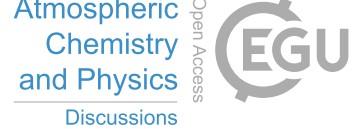

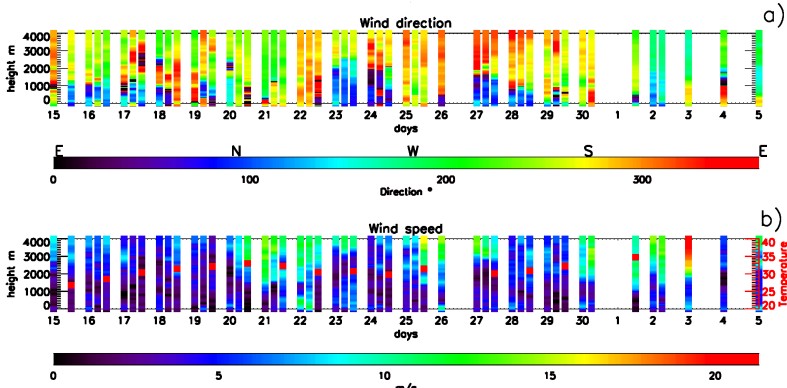

**Figure 3.** Panel a: wind provenience direction. Panel b: wind speed. Ground temperature at 12:00 UTC is also reported as red squares, superimposed to the wind speed profiles.





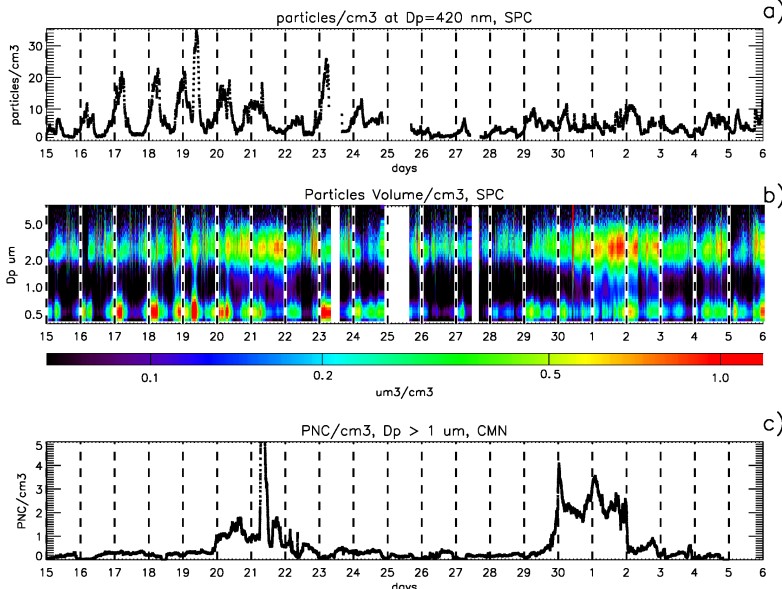

**Figure 4.** Panel a: APSS particle counts at 297–420 nm of diameter. Panel b: time series of the volume size distribution of aerosol particles asobserved by the APSS. The y-axis indicates the volume-equivalent particle diameters in μm while colours report the corresponding volume concentration. Panel c: time series of coarse ($Dp > 1$μm) particle number concentration observed at CMN





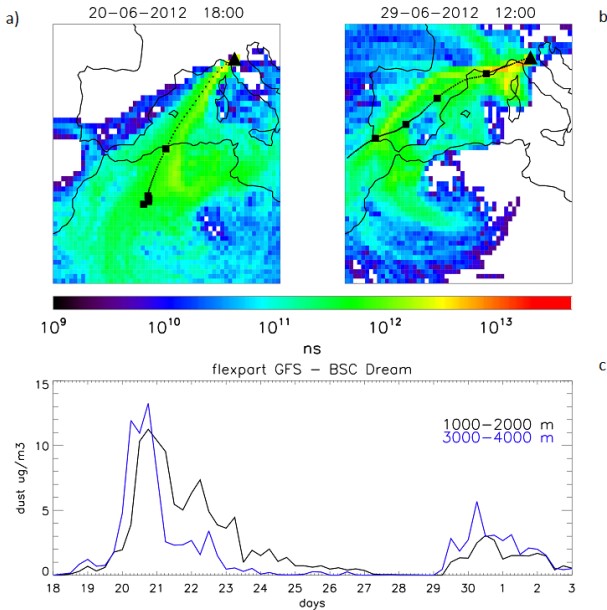

**Figure 5.** FLEXPART backtrajectories over GFS meteorological input: Panels a and b show the footprint (in ns of residence over each bin) of the trajectories released over SPC at 3000 m. Black triangle indicates the point of release, black squares mark the position of the center of mass every 24 hours. The pattern of trajectories released the 20 June at 18:00 UTC are shown on the left (panel a), while pattern released the 29 June at 12:00 UTC on the right (panel b). The simulated dust concentration over the SPC site, derived by the coupling with DREAM, is reported in panel c with a time step of 6 hours. The black line is relative to the particles released at the 1000 – 2000 m atmospheric layer and the blue line to the release at 3000 – 4000 m





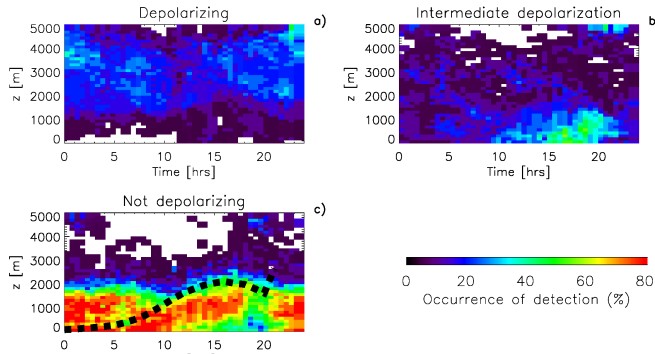

**Figure 6.** Mean diurnal frequency of the vertical distribution of each aerosol class (computed as the ratio between the number of aerosol class detections and the number of days of measurements): Depolarizing (panel a), Intermediate Depolarizing (panel b) and Non-Depolarizing (panel c). The mean PBL height, derived from LiDAR analysis, is reported in black dashed line over the non-depolarizing particles distribution





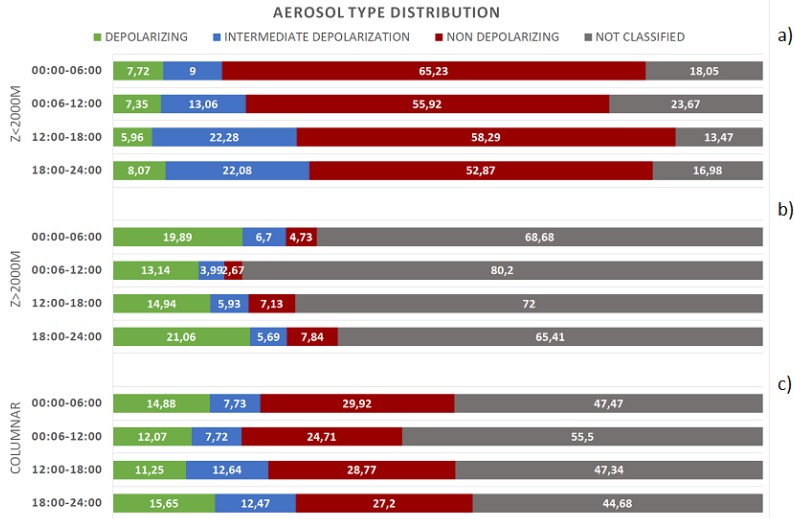

**Figure 7.** Mean diurnal frequency of distribution for the three aerosol classes integrated over the four synoptic intervals (00:00–00:06, 06:00–12:00, 12:00–18:00, 18:00–24:00) below 2000 m (panel a), above 2000 m (panel b) and for the whole atmospheric column 0–5000 m (panel c)



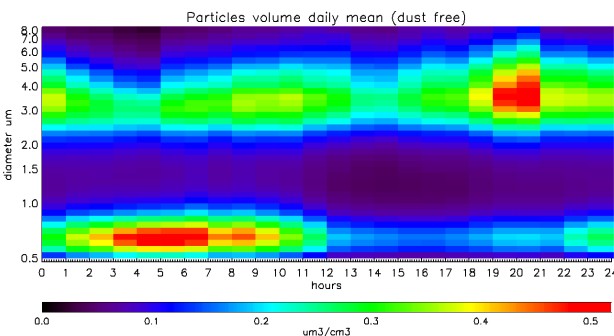

**Figure 8.** Mean diurnal evolution of aerosol particles volume size distribution in dust-free days from the APSS at SPC





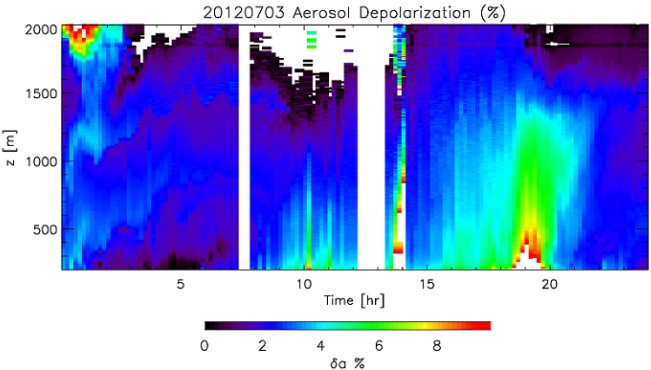

**Figure 9.** Vertical profiles of LiDAR aerosol particles depolarization on the 3 July 2012





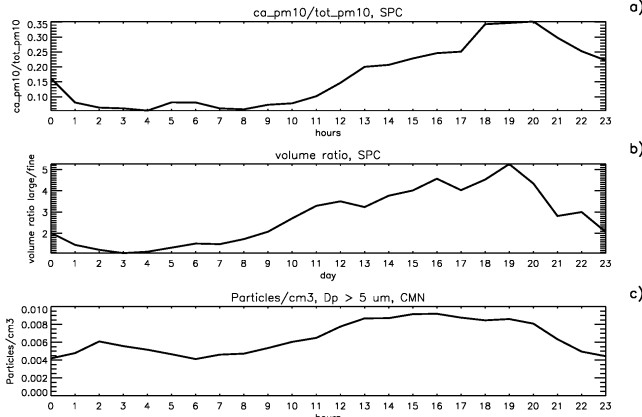

**Figure 10.** Panel a shows the mean diurnal evolution of the ratio of PM10 concentration of the Calcium ion (Ca2+) over the total PM10 ions concentration ( Ca2+PM10 /TotalPM10) in SPC. Panel b reports diurnal mean of the ratio of large particles (1 μm < $Dp$ < 5.5 μm) over the fine ones (0.5 μm < $Dp$ < 1 μm) at SPC while panel c shows the diurnal mean of coarse ($Dp$ >5 μm) particles at CMN. Each mean is computed on dust free days





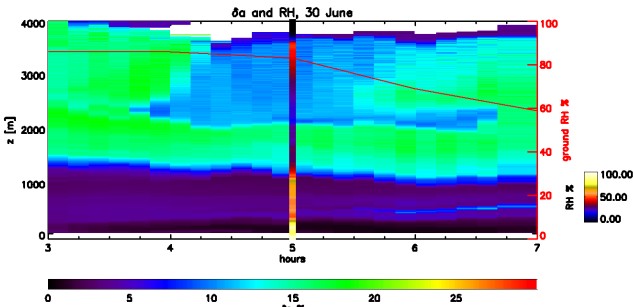

**Figure 11.** Figure reports the vertical profiles of $\delta_a$ and the relative humidity from the radiosounding measurements at 05:00 UTC overlaid with a different color scale; the timeseries of relative humidity observed at the ground is reported as a red line





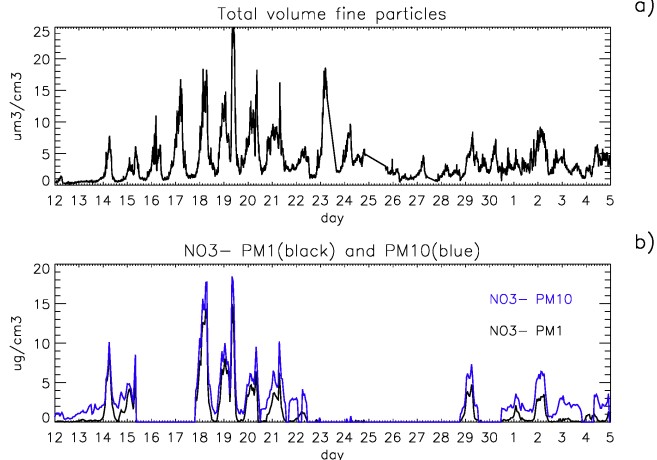

**Figure 12.** APSS fine particles ($Dp < 1\,\mu m$) volume contribution (panel a) compared to nitrates ions concentration (NO3−) both in the PM1 (black) and PM10 (blue) channel (panel b). Zero values corresponds to missing observations