# Peer review of "Vertical distribution of aerosol optical properties in the Po Valley during the 2012 summer campaigns"

_Atmospheric Chemistry and Physics, 2017_

## Referee Comment (RC1) · Anonymous Referee #2 · 29 Jun 2017

**General comments**

In the presented work the authors analyze a 21-day period of atmospheric measurements in the Po Valley, Italy. In particular, the authors present the measurements obtained by different *in situ* and remote sensing instruments and compare them mainly in a qualitative way, presenting the results separately. In this sense, a more quantitative approach should be tried in some analyses. For instance, in the analysis of the effect of RH on the aerosol linear depolarization ratio ($\delta_a$), a correlation analysis between the RH at surface level and the values of $\delta_a$ at the lowest altitudes would offer more information about this process.

According to the authors, "The main objective of this paper is to investigate the transport of desert dust in the middle troposphere and its intrusion into the planetary boundary layer (page 1, line 4)". In order to fulfil this objective it would be desirable a more detailed analysis of the events where the aerosol concentration increases at surface level and how the dust layer interacts with the PBL. However, the whole 21-day period is presented in a single figure (figure 2, lidar; figure 4, in situ).

A great part of this work is based on measurements of a lidar system. However, the lidar signals have not been properly processed. The calculations regarding the lidar measurements must be revised (see specific comments).

Finally, the authors must carry on a thorough revision of the language.

**Specific comments**

**1.** Page 4, line 20: "*In the following discussion, we will make use of backscattering ratio (R)*". Why do the authors use R, which depends on the molecular terms, rather than $\beta_a$, which only depend on aerosols?

**2. Issues regarding the depolarization measurements**

Page 4, from line 21:

"

"

minutes with a vertical resolution of 7.5 m extending, on average, up to 7 km. In the following discussion, we will make use
      of backscattering ratio ($R$) and aerosol depolarization ($\delta a$), defined as (Browell et al., 1990; Cairo et al., 1999):

$$R(r) = \frac{\beta_a(r) + \beta_m(r)}{\beta_m(r)}$$

$$\delta_a = \frac{\beta_{a_{perp}}}{\beta_{a_{par}}}$$

      $\beta_m(r)$ represents the backscatter coefficient from molecular contribution while $\beta_{a_{perp}}$ and $\beta_{a_{par}}$ represent the backscattered
signal components from aerosol light scattering, with polarization respectively perpendicular and parallel to the polarization
      of the emitted light. The inversion of the LiDAR signal is accomplished with the Klett method (Klett, 1985; Fernald , 1984),
      finding a suitable region of the profile that is supposed to be free of aerosol to calibrate the signal, and using piecewise constant
      extinction-to backscattering ratio (LiDAR ratio, $L$) values. We make use of different values for $L$ following what reported
      in literature: desert dust (identified by $\delta_a(r) > 10\%$) is characterized by $L$ equal to 50 sr (Müller et al., 2007) while for low
depolarizing aerosol we assume the values typical for anthropogenic aerosol, L∼60–70 sr (Murayama et al., 1999; Ferrare et
      al., 2001; Fiebig et al., 2002). In addition we considered different values for water cloud ($L = 20$ sr) and ice clouds ($L = 30$ sr)

      (Chen et al., 2002; O'Connor et al., 2004). A more detailed description of the methods used to perform the inversion of LiDAR
      data, together with a through uncertainty analysis, performance in conditions close to the SPC site and additional experimental
      set-up details, are given in Cairo et al. (2012) and Rosati et al. (2016) and its supplementary material.

For the determination of the lidar ratio Rosati [2016] uses the volume linear depolarization ratio, defined as: $\delta \equiv \frac{\beta_{a\perp} + \beta_{m\perp}}{\beta_{a\parallel} + \beta_{m\parallel}}$ (called DR by Rosati [2016]). This parameter indicates the total depolarization effect of the atmosphere (molecules + aerosols), and can be determined in a straightforward way[1] as the calibrated ratio of the elastic signals:

$$\delta = C \frac{P_\perp}{P_\parallel}$$

This parameter can be estimated prior to the signal inversion and can be used as a *rough* indicator of the presence of dust.

Unlike $\delta$, the aerosol depolarization, $\delta_a \equiv \frac{\beta_{a\perp}}{\beta_{a\parallel}}$ (more commonly called aerosol -or particle- linear depolarization ratio), is an intensive property of the aerosols. In order to estimate this parameter we first need to perform the inversion of the signal. Because of this, in a first instance, I could not get to understand how the authors use $\delta_a$ to determine the lidar ratio prior to the signal inversion. However, in page 4, line 29 the authors claim: *"…desert dust (identified by $\delta_a$ (r) > 10%) is characterized by L equal to 50 sr (Müller et al., 2007)…"*. This is the same criterion used by Rosety [2016], although Rosety [2016] used $\delta$ instead of $\delta_a$. Because of this, the most likely explanation is that the authors have mixed up both parameters ($\delta$ and $\delta_a$).

Another example of this is found in page 6, line 29: *"At 532 nm, values of aerosol depolarization around or higher than 30 % are generally associated with layers of nearly pure mineral dust while smaller values (around 8–10 %) are often detected in*
* * *
[1] Strictly speaking we should take into account the optical properties of the system, especially the diattenuation properties and misalignment of the receiving optics [Freudenthaler, 2016; doi:10.5194/amt-9-4181-2016]

*correspondence of mixture of mineral dust and non-depolarizing particles*". Nevertheless, the authors barely find aerosols with $\delta_a > 20$ % (e.g., Fig. 1). This is another indication that the authors are actually using $\delta$ instead of $\delta_a$.

Page 7, line 3: "*The reader should notice that the lower depolarization values that we observe respect what usually found in literature (especially for the dust layers) are more likely linked to the calibration process, and in particular to the difficulty in individuating completely aerosol free layers in the vertical span of the adopted LiDAR system (from ground to 7 Km)*." Again, I think that these differences are because the authors are comparting two different parameters ($\delta$ and $\delta_a$). The authors should revise their calculations to see if this is the source of disagreement. If, after this, the calibration stills play an important role in this disagreement, the authors should describe its effect in a more detailed way.

In the corrected version the authors must indicate, unambiguously, which depolarization parameter are using. The volume depolarization ratio, $\delta$, can still be used as a rough indicator of the presence of dust in the determination of the lidar ratio values. However, for aerosol classification in section 3 I strongly recommend using the aerosol linear depolarization ratio ($\delta_a$), since it is an intrinsic property of aerosols and does not depend on the molecular terms.

**3.** Page 6, line 16. "*For simplicity, dust on emissive areas is considered to be injected uniformly below 1000 m a.g.l., therefore only trajectories crossing this height are included in the footprint-emissions coupling*". Is this a common procedure? Is there any previous work that backs this procedure?

**4.** Page 7, line 5: "*The LiDAR classification, based on the statistical distribution of the overall observed $\delta_a$ and R values, is also applied here to overcome such limitations.*" However, in Fig. 1 it can be seen that the only parameter used for aerosol classification is $\delta$. Because of this, we could get more information with a histogram of only $\delta$ values. Also, the figure should only show relevant $\delta$ values: more than half of the current figure corresponds to $\delta$ values (over 0.25) with almost no associated case.

**5.** Page 7, line 13:

1. low values of $\delta_a$ (< 3 %); based on the above references, these particles may be composed of anthropogenic pollution and, for higher values of $R$, by droplets, and are defined as non-depolarizing.

2. high values of $\delta_a$ (> 10 %); according to the previously mentioned literature, this can be consider as a threshold value for mineral dust or mixed dust particles. This class is defined as depolarizing.

3. intermediate $\delta_a$ values (3 %< $\delta_a$ <10 %) which, based solely on $R$ and $\delta_a$, cannot be considered as indicative of a dominance of a defined aerosol type unless coupled to a more thorough correlation with additional observations. We will refer to this type as intermediate depolarizing.

The authors are comparing their results to other references although they have previously said that their results might not match them due to calibration issues (again, this is likely due to a confusion between $\delta$ and $\delta_a$). If $\delta$ is finally used for the classification (although I strongly recommend $\delta_a$), the threshold values should be derived from a statistical analysis of its values.

**6.** Figure 3. The relevant data in fig. 3 covers less than half of the total area and is hard to visualize. The figure should be redesigned for better interpretation. Also, it could be improved if the surface temperature were plot in an independent panel.

**7**. Page 8, line 29 "*Meteorological evolution is integrated with the aerosol optical variability from LiDAR (see Fig. 2) and with ground aerosol 30 number concentration and volume size distribution (estimated as the volume of a sphere of diameter corresponding to the volume-equivalent particle diameter) at SPC and CMN (see Fig. 4)*" Meteorological parameters, lidar measurements, and in situ measurements are presented separately in different figures. They do not seem to be integrated.

**8.** Page 10, line 18 "*According to FLEXPART, the import of mineral dust persists until the morning of 23 June, when dust presence is not unambiguosly inferable from observations but the aerosol mask still indicates the presence of intermediate depolarizing particles below 2000 m. The second desert dust event predicted by FLEXPART shows the same timing with respect to observations but, while the APSS and the OPSS indicate a similar dust burden for the two desert aerosol advection events, the dust load indicated by the model(between 3 and 5 µg m−3) appears lower respect to the previous events*". In addition to limitations of the model, differences between estimations of FLEXPART over the SPC site at 1-2km and 3-4km and surface-level measurements at SPC (different altitude level) or CMN (different location) might be partly caused by aerosol time-space variability. Have the authors considered a comparison against lidar-derived dust concentration? (These can be retrieved with the POLIPHON method by Mamouri [2014[2]]).

**9.** Figure 7. This figure does not seem to add any information to what can be seen in Fig. 6 and what is on the text. I think it can be removed.

**10. Issues regarding section 7**

Name of the section: "Effect of aerosol hygroscopic growth on aerosol particles light scattering and depolarization" -> Effect of aerosol hygroscopic growth scattering and depolarization

Page 12, line 19. "*LiDAR data (Fig. 2) frequently show, during early morning hours, a shallow layer of non-depolarizing aerosol below 300 m height, more easily visible during days characterized by desert dust and mixed dust events*". Below 300 m the overlap of the lidar is not complete. Despite the authors claim that "*Experimental correction allows the reconstruction of the LiDAR backscattering profile down to around 100 m, with an acceptable uncertainty (4-17)*", they also say that "*The reader should notice that the lower depolarization values that we observe respect what usually found in literature (especially for the dust layers) are more likely linked to the calibration process, and in particular to the difficulty in individuating completely aerosol free 5 layers in the vertical span of the adopted LiDAR system (from ground to 7 Km) (7-5)*" The question is: to what extent can we trust depolarization measurements below complete overlap?

Figure 11 and Figure S2. For this kind of study it might be more appropriate to show the aerosol and humidity profiles at 5:00 UTC as regular plots (not color plots).
* * *
[2] doi:10.5194/amt-7-3717-2014

Page 12, line 29: "*The study, extended separatedly on the whole dust events 30 period (20–23 June and 29 June – 2 July, lower panel of Fig. S2, supplementary material) and in the remaining dust free days (upper panel of Fig S2, supplementary material), indicates in both case a depolarization decrease in the lower layers, visible starting from RH>60%*". In Fig. 11 and Fig. S2 we can see that in cases with presence of dust the depolarization decreases with RH. But this can be seen as the opposite: elevated dust layers (within dry hot air-masses from the Sahara) uncoupled from the PBL result in a decrease in the RH. In order to state that the dust depolarization properties are affected by the humidity, the authors should prove that a noticeable amount of dust actually reaches the lower altitudes where RH > 60 %. For instance, for the case used as example (30 June), at 5 UTC (time of the sounding) we do not see an especially high concentration of coarse particles at surface level compared to, for instance, 1 July (Fig. 4). Also, it would be interesting if the authors showed, in addition to the RH profile, the temperature profile for 30 June. This way we could see it a thermal inversion between the dust and the lower layers keeps them uncoupled. Finally, on 1 July we can see aerosols classified as dust reaching the surface (it is also the day with highest coarse-mode concentration at surface level). Although no soundings might be available at those times, it would be interesting to compare the depolarization values at the lowest altitudes available and the relative humidity at surface level.

**Other comments, language errors, and typos**

Although some language mistakes have been noticed, the following list is not complete. Because of this, a thorough revision of the language must be made.

- Common language error (1): Before a certain characteristic (e.g., concentration, depolarization, size…) the noun is usually singular (e.g., particle size, aerosol depolarization, ion concentration), not plural (e.g., particles size, aerosols depolarization, ions concentration).
- Common language error (2): Before a date (e.g., 30 June) you do not have to write "the" (e.g., the 30 June).
- $\delta_a$ should be called aerosol (or particle) linear depolarization ratio instead of the ambiguous "Aerosol depolarization". $\delta$ should be called Volume linear depolarization ratio
- 4-7: "For the most part of the year". -> For the greater part of the year
- The lidar ratio is referred as both L (e.g. 4-28) and LR (e.g., 6-24). This has to be fixed.
- 5-29 "High accuracies" -> high accuracy.
- 8-29 "small particles concentration" -> small particle concentration (see common language error (1)).
- Figure 2: "not depolarizing aerosol (yellow), depolarizing aerosol (orange) and intermediate aerosol (brown) properties". Rewrite this phrase so that it makes sense.
- 9-19 "*…intensification of mineral dust burden or, as suggested by ??? a corresponding increase in black carbon concentration observed at CMN (see also Cristofanelli et al. (2016)), by mixing with pollution from the regional PBL (Cristofanelli et al., 2009).*" Rewrite to make it more clear.
- Figure 4. Title of panel a): "Dp = 420 nm" -> 297nm< Dp < 420 nm
- 10-19 "unambiguosly" -> unambiguously
- Figure 8. "…evolution of aerosol particles volume size distribution" -> …evolution of particle volume-size distribution (see common language error (1)).
- Figure 10. "… LiDAR aerosol particles depolarization …" .-> volume linear depolarization ratio (in case of $\delta$) or particle linear depolarization ratio (in case of $\delta_a$)
- 11-33 "wide range of aerosol type" -> wide range of aerosol types
- Figure 10. The label of the horizontal axes should be "hour" in all panels not "hours" or "day".
- 12-20. "*see for instance 00:00-06:00 UTC of the 19 June and between 00:00-08:00 UTC of 30 June*" -> see for instance 00:00-06:00 UTC on 19 June and between 00:00-08:00 UTC on 30 June (see common language error (2)).
- 12-31. "*…indicates in both case a depolarization decrease in the lower layers, visible starting from RH>60%.*" -> …indicates in both cases a decrease in the depolarization of the lower layers for RH>60%.
- 14-27 "*…basing on in situ measurements…*" -> based on *in situ* measurements

---

## Referee Comment (RC2) · Anonymous Referee #3 · 3 Oct 2017

This study investigates the transport of desert dust in the middle troposphere and its intrusion into the planetary boundary layer (PBL) based on the field campaign over the Po Valley. In situ and remote sensing measurement results were exhibited together with Lagrangian air masses transport simulations to explain the effects of meteorological evolution and transport patterns on the aerosol variability. However, the main text of the manuscript needs to be more logically organized, and it is suggested to modify the logical structure of some sections to give clearer conclusions.

For instance, the whole introduction part is in one large paragraph mentioning the characteristics of the Po river basin, previous studies over the Po Valley, mineral dust's adverse impact, and LiDAR observations, which is a little unclear. It is recommended to re-organize some of the sentences or separate this part into more than one paragraph,

and provide a clearer logical sequence introducing your study focus.

For the conclusion part, it is recommended to highlight a few key findings or conclusions using refined expression of the evidence.

Finally, some language mistakes have been noticed and revision of the language is needed to give clearer meaning of the sentences. For instance: Page 3, line 15: '. . .basing on the analyses of continuous and vertically resolved particles light scattering and depolarization. . .' -> consider revising
* * *

---

## Author Comment (AC1) · 22 Dec 2017

We would like to thanks the reviewers for the useful comments and insights that allowed us to better redefine the methods and presenting the results of this study.
First reviewer:

**General comments**

" [. . .]in the analysis of the effect of RH on the aerosol linear depolarization ratio ($\delta_a$), a correlation analysis between the RH at surface level and the values of $\delta_a$ at the lowest altitudes would offer more information about this process"

[Figure]

Following the suggestions of the reviewer, we adopted a more quantitative approach to describe the near ground hygroscopic growth events. A correlation analysis between the ground RH and the $\delta_a$ along the time would not be meaningful, as the variability of aerosol depolarization near the ground is not affected solely by RH. We decided instead to better show, along the whole campaign period, the effect of different values of RH along the vertical profile of $\delta_a$ for dust and dust-free days. Details can be found in the answer to point "10. Issues regarding section 7"

" *According to the authors, "The main objective of this paper is to investigate the transport of desert dust in the middle troposphere and its intrusion into the planetary boundary layer (page 1, line 4)". In order to fulfil this objective it would be desirable a more detailed analysis of the events where the aerosol concentration increases at surface level and how the dust layer interacts with the PBL. However, the whole 21-day period is presented in a single figure (figure 2, lidar; figure 4, in situ).* "

To facilitate the individuation of dust intrusion in the PBL and its penetration down to the ground, a closer view is added (see Fig 5 of the revised manuscript). Here we show LiDAR aerosol particle types profiles, and the corresponding APSS volume size distribution of aerosol particles at the ground, during the dust events. The interaction of dust with the PBL and the direct effect of the dust layer intrusion on the particles concentration at the ground are more easily identifiable.

" *A great part of this work is based on measurements of a lidar system. However, the lidar signals have not been properly processed. The calculations regarding the lidar measurements must be revised (see specific comments).* "

Reviewer comments highlighted the need to revise the section on the adopted LiDAR signal processing. We rewrote this section to remove any ambiguities in the parameters definition. See also answers to points 2, 4 and 5.

" *Finally, the authors must carry on a thorough revision of the language.* "

We revised the manuscript to improve the language.

**Specific comments**

" *1. Page 4, line 20: "In the following discussion, we will make use of backscattering ratio (R)". Why do the authors use R, which depends on the molecular terms, rather than $\beta_a$, which only depend on aerosols?* "

The use of the total backscattering ratio R, instead of the aerosol backscatter $\beta_a$, has the advantage to be bereft of the noise that is introduced by the signal inversion for $\beta_a$ estimation. This helps to have a refined aerosol classification. In the paper we therefore based the aerosol classification on the parameter 1-1/R, varying from 0 to 1 accordingly to the presence of an aerosol layer, instead of relying directly on R values. To emphasize the meaning of such parameter we changed its expression as a function of the aerosol backscattering ratio (Ra=R-1= $\beta_a$(r)/ $\beta_m$(r)) instead of R. Classification is now a function of the values of Ra/(Ra+1), still varying from 0, when $\beta_a$(r)=0, to 1, when in presence of dense aerosol layer (high $\beta_a$(r) values).

" *2. [...] I could not get to understand how the authors use $\delta_a$ to determine the lidar ratio prior to the signal inversion. However, in page 4, line 29 the authors claim: "...desert dust (identified by $\delta_a$ (r) > 10%) is characterized by L equal to 50 sr (Müller et al., 2007)...". This is the same criterion used by Rosety [2016], although Rosety [2016] used $\delta$ instead of $\delta_a$. Because of this, the most likely explanation is that the authors have mixed up both parameters ($\delta$ and $\delta_a$). Another example of this is found in page 6, line 29 [...] Nevertheless, the authors barely find aerosols with $\delta_a$ > 20 % (e.g., Fig. 1). This is another indication that the authors are actually using $\delta$ instead of $\delta_a$. Page 7, line 3: [...] Again, I think that these differences are because*

[Figure]

*the authors are comparting two different parameters ($\delta$ and $\delta_a$). The authors should revise their calculations to see if this is the source of disagreement. If, after this, the calibration stills play an important role in this disagreement, the authors should describe its effect in a more detailed way. In the corrected version the authors must indicate, unambiguously, which depolarization parameter are using. The volume depolarization ratio, $\delta$, can still be used as a rough indicator of the presence of dust in the determination of the lidar ratio values. However, for aerosol classification in section 3 I strongly recommend using the aerosol linear depolarization ratio ($\delta_a$), since it is an intrinsic property of aerosols and does not depend on the molecular terms. "*

The reviewer correctly stresses a lack of clarity in the text, where in fact the parameters seem to be mixed. It is true that the aerosol depolarization can only be ascertained after the signal has been processed and then inverted in term of R, so that it cannot be used during the inversion. In our case the inversion of the LiDAR signal is accomplished with the Klett method using piecewise constant extinction to backscatter ratio (LR) values, chosen accordingly to the volume depolarization (DR, in Rosati et al., 2016) and not aerosol depolarization, as would be proper to correctly discriminate among different classes of aerosol. However, when the aerosol load is significant – and these are the cases where an improper choice of LR would have more effect - the value of the volume tends to that of the aerosol depolarization, so that the use of the volume as a proxy of the aerosol in the inversion of the LiDAR signal can be accepted. We do not think it is worthwhile to dwell too much on that in the text, but we need to remove the ambiguity in the text pointed out by the reviewer. To this aim we would then rewrite it as: (from 4,29 on) "... following the values reported in literature: using Volume depolarization as proxy of aerosol depolarization, highly depolarizing desert dust ($\delta$ (r) <10) is characterised by L...".

On the contrary, on page 6, line 34 we really meant Aerosol depolarization.

As the reviewer suggests, we expanded the discussion on inaccuracies in depolarization in (7,3...) as follows: "The reader should notice that the lower depolarization values that we observe respect what usually found in literature (especially for the dust layers) are likely linked to the calibration process, and in particular to the difficulty in individuating completely aerosol free layers in the vertical span of the adopted LiDAR system (from ground to 7 Km). In this work, depolarization has been calibrated following the "0° calibration" or the "atmospheric calibration" procedure, i.e. making use of a low aerosol height range in the lidar signal, where only the molecular contribution could be considered. There, the volume depolarization ratio has been forced to assume the well-known value of the air molecule linear depolarization ratio (Behrendt and Nakamura, 2002). We acknowledge that this calibration is unsatisfactory to produce quantitative results, as the possible residual presence of small amounts of highly depolarizing aerosol in the assumed clean range can easily compress the range of variability of the volume depolarization, and underestimate the final depolarization products (Freudenthaler et al., 2009; Freudenthaler, 2016). However, this possible source of inaccuracy does not compromise the purpose of this work."

" *3. Page 6, line 16. "For simplicity, dust on emissive areas is considered to be injected uniformly below 1000 m a.g.l., therefore only trajectories crossing this height are included in the footprint-emissions coupling". Is this a common procedure? Is there any previous work that backs this procedure? "*

The choice of 1000m as a level for mineral dust injection is indeed a sharp assumption. We decided therefore to estimate the level of injection as the PBL top height, extracted by FLEXPART itself from the GFS input meteorological field (Stohl et al. 2010), instead of adopting a fixed height. This does not significantly change the results of the analysis (see also answer to point 8)

" *4. Page 7, line 5: "The LiDAR classification, based on the statistical distribution of the overall observed $\delta_a$ and R values, is also applied here to overcome such limitations." However, in Fig. 1 it can be seen that the only parameter used for aerosol classification is $\delta$. Because of this, we could get more information with a histogram of only $\delta$ values. Also, the figure should only show relevant $\delta$ values: more than half of the current figure corresponds to $\delta$ values (over 0.25) with almost no associated case.* "

" *5. Page 7, line 13: [. . .] The authors are comparing their results to other references although they have previously said that their results might not match them due to calibration issues (again, this is likely due to a confusion between $\delta$ and $\delta_a$). If $\delta$ is finally used for the classification (although I strongly recommend $\delta_a$), the threshold values should be derived from a statistical analysis of its value* "

The distribution of $\delta_a$ and 1-1/R values (now Ra/(1+Ra)) is indeed what is really driving the aerosol classification. We agree with the review that was not clearly presented in the text and in Fig.1 of the manuscript. We corrected the text (from page 7, line 1: "Here we implement a three-types aerosol discrimination scheme to characterize the vertical and temporal aerosol variability over the region along the campaign period (15 June 2012 – 5 July 2012) based on the different statistical distribution of optical properties of each class (see Fig. 1). The reader should notice that [. . .]. The LiDAR classification, based on the statistical distribution of the overall observed $\delta_a$ and $Ra$ values, is in fact applied here to overcome such limitations. The robustness of the results is then further supported by comparison with Lagrangian analysis and in-situ measurements.") and modified the axes of Fig.1. to magnify the region on which the values are mostly concentrated. It's easier now to identify the three different distributions of optical properties and the resulting classification. As is possible to see, each class shows a different range on the Ra/(1+Ra) parameter. We restricted therefore the threshold for

classification on such intervals.

From (7,12): "The different aerosol classes can be discerned in three distinct patterns:

1. $0.1 < Ra/(Ra+1) < 0.8$ and low values of $\delta_a$ (< 3 %); such low values of aerosol linear depolarization ratio are indicative of spherical particles. These particles may be composed of anthropogenic pollution and, for higher values of $Ra$, by droplets, and are defined as non-depolarizing.

2. $0.3 < Ra/(Ra+1) < 0.7$ and high values of $\delta_a$ (> 10 %); In this class we find the highest values of aerosol linear depolarization ratio (mainly ranging from 10% to 20%) and this can be indicative of the presence of mineral dust particles. This class is defined as depolarizing.

3. $0.2 < Ra/(Ra+1) < 0.6$ and intermediate $\delta_a$ values (3 %< $\delta_a$ <10 %) which, based solely on $Ra$ and $\delta_a$, cannot be considered as indicative of a dominance of a defined aerosol type unless coupled to a more thorough correlation with additional observations. We will refer to this type as intermediate depolarizing.

"

As expected, the resulting aerosol mask (see Fig.2 of the revised manuscript) appears to be not meaningfully affected by such modifications.

" *6. Figure 3. The relevant data in fig. 3 covers less than half of the total area and is hard to visualize. The figure should be redesigned for better interpretation. Also, it could be improved if the surface temperature were plot in an independent panel.* "

As the purpose is to emphasize the evolution of meteorological conditions during the different phases of transport, we believe that it would be useful to represent the whole

period in the same figure. We increased nevertheless the plot dimensions, to facilitate the visualization of the main phases (as the dust transport events, that should be visible as variation in wind directions and increased wind intensity respect to the other days). Ground temperature is reported on a separate panel to improve the clarity of the figure.

" *7. Page 8, line 29 "Meteorological evolution is integrated with the aerosol optical variability from LiDAR (see Fig. 2) and with ground aerosol 30 number concentration and volume size distribution (estimated as the volume of a sphere of diameter corresponding to the volumeequivalent particle diameter) at SPC and CMN (see Fig. 4)" Meteorological parameters, lidar measurements, and in situ measurements are presented separately in different figures. They do not seem to be integrated. "*

What we meant here is that the results are derived from the addition and comparison of different information. To avoid confusion we changed the terminology from "Integrated" to "compared".

" *8. Page 10, line 18 "According to FLEXPART, the import of mineral dust persists until the morning of 23 June, when dust presence is not unambiguously inferable from observations but the aerosol mask still indicates the presence of intermediate depolarizing particles below 2000 m. The second desert dust event predicted by FLEXPART shows the same timing with respect to observations but, while the APSS and the OPSS indicate a similar dust burden for the two desert aerosol advection events, the dust load indicated by the model(between 3 and 5 $\mu gm^{-3}$) appears lower respect to the previous events". In addition to limitations of the model, differences between estimations of FLEXPART over the SPC site at 1-2km and 3- 4km and surface-level measurements at SPC (different altitude level) or CMN (different location) might be partly caused by aerosol time-space variability. Have the authors considered a comparison against lidar-derived dust concentration? (These can be retrieved with the POLIPHON method by*

*Mamouri [20142]). "*

The use of FLEXPART trajectories in this paper is serving as a support for assessing the nature of the observed particle layers. It is out of the scope of the study to provide an accurate estimate of the amount of dust transported, that would require indeed to take into account several possible sources of variability. Moreover, the possible inaccuracies on the aerosol depolarization values retrieval presented above, make difficult to apply the POLIPHON method properly. We believe that this would lead to an additional work that would go far beyond the purpose of the paper. We decided therefore to report dust air mass fraction evolution instead of the mineral dust reconstructed concentration, to avoid additional uncertainties coming from the estimate of the aerosol load, as rightly suggested by the reviewer.

The information brought by such analysis, ancillary to the understanding of the nature of the layers observed by the LiDAR and in-situ measurements, remains mostly unchanged. The text on section 2.5 will be modified as follows: "To give an estimate of the variability in the mass of mineral dust advected over SPC we compute, for each release, the mass fraction of trajectories that encounter dust emissive regions respect to the total mass of the released cluster").

From (10,16): "Maximum mineral aerosol fraction from FLEXPART analysis occurs on June 20, both at the upper layer (9%) and at the bottom layer (9% also. According to FLEXPART, the import of mineral dust at the lower layer persists until the morning of 23 June, when dust presence is not unambiguously inferable from observations but the aerosol mask still indicates the presence of intermediate depolarizing particles below 2000 m. The second desert dust event predicted by FLEXPART again shows the same timing with respect to observation and also confirms the presence of a thick layer of dust that involves at the same time the 1000–2000 m and 3000–4000 m layers. The estimated mass fraction contribution on the contrary, especially in the lower layer (between 2 and 4 %), appears inferior respect to the previous events."

" *9. Figure 7. This figure does not seem to add any information to what can be seen in Fig. 6 and what is on the text. I think it can be removed.* "

We agree that this information may be redundant, and we removed it from the manuscript.

" *10 .Issues regarding section 7:*

*Name of the section: "Effect of aerosol hygroscopic growth on aerosol particles light scattering and depolarization" -> Effect of aerosol hygroscopic growth scattering and depolarization* "

Name of the section modified.

"*Page 12, line 19. "LiDAR data (Fig. 2) frequently show, during early morning hours, a shallow layer of non-depolarizing aerosol below 300 m height, more easily visible during days characterized by desert dust and mixed dust events". Below 300 m the overlap of the lidar is not complete. Despite the authors claim that "Experimental correction allows the reconstruction of the LiDAR backscattering profile down to around 100 m, with an acceptable uncertainty (4-17)", they also say that "The reader should notice that the lower depolarization values that we observe respect what usually found in literature (especially for the dust layers) are more likely linked to the calibration process, and in particular to the difficulty in individuating completely aerosol free 5 layers in the vertical span of the adopted LiDAR system (from ground to 7 Km) (7-5)" The question is: to what extent can we trust depolarization measurements below complete overlap?*"

For what concerns the uncertainties on the depolarization in the region of partial overlap, the Volume depolarization $\delta$ (being derived as a ratio of two signals both characterized by the same partial overlap loss) is not affected. Not so the $\delta_a$, which is computed from $\delta$ and the total backscatter ratio (R), which can be impacted by systematic errors when retrieved in the partial overlap region. Errors on the R are discussed in Biavati et al. (2011), where the algorithm for partial overlap correction was presented; for the present work, the relative error on the reconstructed R remains below 10% in the altitude range where the correction has been applied.

The relative error on the $\delta_a$ in the region of partial overlap decrease with increasing R and increase for increasing $\delta$ . For the values of $\delta$ that we found in our work, the error on $\delta_a$ remain below 15% for R greater than 1.5 and below 30% for R greater than 1.25.

"*Figure 11 and Figure S2. For this kind of study it might be more appropriate to show the aerosol and humidity profiles at 5:00 UTC as regular plots (not color plots).*"

We modified the plots accordingly.

"*Page 12, line 29: [. . .]. In Fig. 11 and Fig. S2 we can see that in cases with presence of dust the depolarization decreases with RH. But this can be seen as the opposite: elevated dust layers (within dry hot airmasses from the Sahara) uncoupled from the PBL result in a decrease in the RH. In order to state that the dust depolarization properties are affected by the humidity, the authors should prove that a noticeable amount of dust actually reaches the lower altitudes where RH > 60 %. For instance, for the case used as example (30 June), at 5 UTC (time of the sounding) we do not see an especially high concentration of coarse particles at surface level compared to, for instance, 1 July (Fig. 4). Also, it would be interesting if the authors showed, in*

*addition to the RH profile, the temperature profile for 30 June. This way we could see it a thermal inversion between the dust and the lower layers keeps them uncoupled. Finally, on 1 July we can see aerosols classified as dust reaching the surface (it is also the day with highest coarse-mode concentration at surface level). Although no soundings might be available at those times, it would be interesting to compare the depolarization values at the lowest altitudes available and the relative humidity at surface level. "*

We used the 30 June as a case example as in this day the effect of RH variation on the vertical profile of $\delta_a$ was particularly easy to individuate. In figure 1 we compare the $\delta_a$ and RH profiles with the potential temperature (TH) profile: in particular, in contrast to the nearly isothermal layer in correspondence of the core of the dust layer, we can notice that the $\delta_a$ depletion, and the RH sharp increase observed below 500m, are associated to a notably more stable TH profile.

In this case, nevertheless, we agree with the reviewer that the amount of dust at the ground was less meaningful than what observed during the 1 July, but indeed for this day we don't have the early morning radiosounding. We therefore report in figure 2 the temporal evolution of $\delta_a$ near ground ($\sim$150m) compared with RH and Temperature at the ground. It is possible to notice the clear decrease of $\delta_a$ during the early hours of the morning, associated to high values of RH (higher than 70%) and low values of temperature (less than 25°C). During late afternoon this effect is less evident both because of less extreme values of RH and T and because of the deposition of coarser and more depolarizing particles (see for example Fig. 2 and Fig.4 of the manuscript).

In the manuscript, for sake of completeness, we decided then to remove the single case study and discuss the vertical profiles of $\delta_a$, TH and RH for dust and dust free days, where dust days corresponds to the days of increased coarse particles at the ground (from 20 to 22 June and from 30 June to 2 July). We discuss it on section 7, from (12,23) on.

**Other comments, language errors and typos:**
The manuscript was revised to fix language mistakes and typos.

**References:**

Behrendt A. and Nakamura T., 2002: Calculation of the calibration constant of polarization lidar and its dependency on atmospheric temperature. Opt. Express 10, 805-817.

Biavati, G., Donfrancesco, G., Cairo, F. and Feist, D.: Correction scheme for close-range lidar returns, Appl. Opt. 50, 5872-5882, 2011.

Freudenthaler, V., Esselborn, M., Wiegner, M., Heese, B., Tesche, M., Ansmann, A., Müller, D., Althausen, D., Wirth, M., Fix, A., Ehret, G., Knippertz, P., Toledano, C., Gasteiger, J., Garhammer, M., and Seefeldner, M.: Depolarization ratio profiling at several wavelengths in pure Saharan dust during SAMUM 2006, Tellus B, 61, 165–179, 2009.

Freudenthaler, V.: About the effects of polarising optics on lidar signals and the $\Delta 90$ calibration, Atmos. Meas. Tech., 9, 4181-4255, https://doi.org/10.5194/amt-9-4181-2016, 2016.

Stohl, A., Sodemann, H., Eckhardt, S., Frank, A., Seibert, P., Wotawa, G., Morton, D., Arnold, D. and Harustak, M.: The Lagrangian particle dispersion model FLEXPART version 9.3, Tech. rep., Norwegian Institute of Air Research (NILU), Kjeller, Norway,

available at: http://flexpart.eu, last access: 2 June 2016, 2010.

[Figure]

[Figure]

**Fig. 1.** Vertical profiles of delta_a (black), TH (red) and RH (blue) for 30 June, 05:00 UTC.

[Figure]

**Fig. 2.** Temporal evolution of delta_a (black) at 150m, and TH (red) and RH (blue) at the ground, 1 July case study

---

## Author Comment (AC2) · 22 Dec 2017

We would like to thanks the reviewers for the useful comments and insights that allowed us to better redefine the methods and presenting the results of this study.

Second reviewer:
" *[. . .]However, the main text of the manuscript needs to be more logically organized, and it is suggested to modify the logical structure of some sections to give clearer conclusions. For instance, the whole introduction part is in one large paragraph mentioning the characteristics of the Po river basin, previous studies over the Po Valley, mineral dust's adverse impact, and LiDAR observations, which is a little unclear.*

*It is recommended to re-organize some of the sentences or separate this part into more than one paragraph and provide a clearer logical sequence introducing your study focus. For the conclusion part, it is recommended to highlight a few key findings or conclusions using refined expression of the evidence.*"

We thanks the reviewer for the comment. We re-organized the manuscript introduction to have a clearer logical structure (presentation of the characteristics of the Po Valley, description on the possible sources of particulate over the region, goals, methods and focus of the study) and we introduced a sub-paragraph structure to separate the description of anthropogenic and natural PM sources.

Similarly, in the conclusions, we stressed the results in a clearer form. In particular:

We removed lines from (14,7) to (14,11).

From (14,21): "...carrying depolarizing aerosol.The study offered evidence of dust transport to the ground, showing clear dust layers intrusion in the PBL and rapid mixing with local pollution. We showed how this mixed layer, generally characterized by lower depolarization values, can reach the ground within few hours and we showed, by direct comparison with ground in situ instruments, the corresponding enhancement of particle volume size distribution in the 2-5 μm range (leading to values higher than 1 $\mu m^3 cm^{-3}$)."

We substituted lines (14,30-31) with "The study revealed moreover the presence of events of late afternoon particles resuspension from the soil, not related to Saharan dust transport, impacting on the PM concentration near the ground. The existence of..."

From (15,6): "The combination of depolarization profiles with meteorological and aerosol measurements allowed also to highlight the effects of the condition of high RH (typical for this region) on the particle processes. The analysis revealed how, in correspondence of a shallow layer near the ground (<500 m), in conditions of high relative humidity values ($RH > 60\%$), the aerosol linear depolarization ratio decreases respect to the above layer. Such effect is particularly visible when in presence of mineral dust particles near the ground ($\delta_a$ decrease $\sim$3.5 %). The temporal evolution and the high values of nitrates ion concentration (up to 18 µg m$^{-3}$) in the PM1 and PM10 channels during this period, suggest that the origin of such low depolarization particles can be related to processes of secondary organic aerosol formation and hygroscopic growth on mineral dust particles with nitrate-enriched surface."

" *Finally, some language mistakes have been noticed and revision of the language is needed to give clearer meaning of the sentences. For instance: Page 3, line 15: '. . .basing on the analyses of continuous and vertically resolved particles light scattering and depolarization. . .' -> consider revising*"

We performed a revision of the manuscript to improve the clarity of the text as tracked in the revised version.
* * *

---

## Author Response (AR2)

**Answers to the reviewers**

We thanks the reviewer for the useful remarks that helped improve the results presentation.

*"P7-L10: "the well-known value of the air molecule linear depolarization ratio (Behrendt and Nakamura , 2002)".*
*Behrendt and Nakamura [2002] describe the method to estimate the linear depolarization ratio of air molecules ($\delta_{mol}$). Never-*
*theless, the value of this parameter depends on air temperature, laser wavelength and, mainly, on the shape and bandwidth of*
*the optical filters used by the polarized channels of the lidar. Therefore, the value is not "well-known" but it has to be estimated*
*taking into account the properties of the lidar system.*
*I agree with the authors that the absolute values of aerosol linear depolarization ratio do not affect the main purpose of this*
*parameter, which is to classify aerosols. Nevertheless it would be desirable that the authors specified the value of $\delta_{mol}$ used*
*for the calibration."*

We corrected the text from Page 7,line 10:

"There, the volume depolarization ratio has been forced to assume a reference value for the air molecule linear depolarization
ratio derived from literature, in our case $\delta_{mol}$=1.3 %, given the instrumental setup of the LiDAR and the measurement condi-
tions, see Behrendt A. and Nakamura T. (2002))"

This value corresponds to a full widths at half maximum (FWHM) of the receiver of 10 µm.

*"Section 7: Figure 11 shows the mean $\delta_a$, RH and TH profiles obtained during dust-free and dust days. In the right panel it*
*can be observed that high RH values lead to low $\delta_a$ values close to the surface. However, there are some problems using this*
*figure to infer the hygroscopic growth of dust (as pointed out in the revision). The first one is that dust layers arrive several*
*kilometers above ground level. Therefore, it is no surprise to observe the highest mean $\delta_a$ values at higher altitudes (where*
*dust is present during most of the time) rather than at lower altitudes, regardless of the values of RH."*

The entire profile, from the ground to 4000 m, was presented to give a complete view on the stratification of the atmosphere,
easily comparable with the LiDAR mask of Fig.2. Indeed, the process of hygroscopic growth is happening mainly from 500
m down to the ground, where the decrease in the depolarization is instead clearly corresponding with the sharp increase in the
RH profile, but still with enhanced presence of coarse aerosol at the ground, as seen from the APSS (see answer below).

*"And second, if we assume that there is hygroscopic growth of the dust, and that this growth leads to a decrease in $\delta_a$,*
*then this parameter cannot be used to confirm the presence of dust in the lower altitudes. The presence of dust in the lower*

*altitudes can be confirmed with measurements of the APSS, provided that this system measures the aerosols in dry conditions (and thus large particles due to hygroscopic growth are not confused with dust particles). If this is the case, authors should add a figure comparing the coarse mode fraction, the relative humidity at surface level and $\delta_a$ at the lower altitudes during the dust events (similar to Figure 2 in the author response but adding the measurements from the APSS). This figure will add*
5     *valuable information that will allow the authors to confirm or review their conclusions."*

    The observation of the reviewer is correct. The new figure 11 of the manuscript was meant indeed to resolve such uncertainties. The dust days are in fact selected basing on the detection of enhanced coarse aerosol at the ground from the APSS (not affected by RH). We agree that this was not clarified in the text.

10     Taking into account the two above-discussed points, we hence separated the upper and lowermost layer discussion and added, from page 13, line 1:

    "The average profiles of $\delta_a$, relative humidity ($RH$) and potential temperature ($TH$) are reported in Fig. 11 along the whole range of LiDAR observations. The dust days are selected in correspondence of enhanced presence of coarse aerosol at the
15     ground as seen from the APSS, i.e. when for Dp > 2 μm the volume size distribution reaches values higher than 0.5 μm$^3$ cm$^{-3}$ (20 –22 June and 30 June – 2 July, see also Fig.5 ).
During dust-free days [...]
Conversely, the effect of possible phenomena of hygroscopic growth starts to be visible in the lowermost troposphere (below 500 m). Such layer, marked by the increased stability shown by the $TH$ profile, is characterized by a sharp decrease in $\delta_a$..."

20     We believe that this will answer the reviewer objection while, as mentioned in the interactive discussion phase, a correlation analysis between the ground RH and the $\delta_a$ along the time would not be completely meaningful, as the variability of aerosol depolarization near the ground is not affected solely by RH and it could be difficult to disentangle the different effects.

    *Language:*
*"P3-L38: ". . . compared to the urban one. . . " -> ". . . compared to the urban ones. . . "*
25     *P9-L31: ". . . as seem to be suggested by the observed corresponding. . . " -> as suggested by the observed corresponding*
*In some cases the word diurnal is used as daily, although they do not mean the same. Diurnal makes reference only to those things happening during daylight hours. Check it.*
*In the expressions "above/below X m height" the word "height" should be removed. "*

    The manuscript was revised accordingly.

30     *Figures:*
*Fig. 2, 3 and 4. For better reading, indicate in the figures the periods classified as dust.*

*Fig. 3. Panel a). For a better visualization of the wind direction, use a color bar in which both extremes have the same color.*

*Fig. 6. Label the panels.*

Figures where modified following the suggestions of the reviewer.

**References**

[revised manuscript text omitted]